# Feature Distillation Improves Zero-Shot Transfer from Synthetic Images

**Niclas Popp**
*Bosch Center for Artificial Intelligence, Robert Bosch GmbH, University of Tübingen*
*niclas.popp@de.bosch.com*

**Jan Hendrik Metzen**
*Bosch Center for Artificial Intelligence, Robert Bosch GmbH*

**Matthias Hein**
*University of Tübingen*
*matthias.hein@uni-tuebingen.de*

**Reviewed on OpenReview:** *https://openreview.net/forum?id=SP8DLl6jgb*

## Abstract

Vision-language foundation models such as CLIP have showcased impressive zero-shot capabilities. However, their applicability in resource-constrained environments is limited due to their size and the resulting latency. Knowledge distillation allows to mitigate these challenges by distilling small image encoders that can replace the large CLIP image encoder. In a zero-shot setting, where only the class names are known, no real domain images can be used for this process. Instead, we investigate the use of synthetic images for this purpose. Unlike existing works that focus on improving the quality of synthetic images to bridge the performance gap compared to training on natural images, we find the choice of loss to be a crucial factor. Specifically, minimizing only the distance between the student and teacher image features, without incorporating image captions in the loss function, increases the robustness to spurious features and data corruptions. As a result, this feature distillation approach greatly improves the transfer performance from synthetic to real images. Leveraging these insights, we are able to train domain-specific students that achieve zero-shot performance comparable to a ViT-B/32 teacher on six fine-grained classification datasets while using up to 92% fewer parameters.

## 1 Introduction

**Motivation.** Image classifiers built on top of large vision(-language) foundation models, such as CLIP (Radford et al., 2021) or DINOv2 (Oquab et al., 2023), have shown impressive zero-shot capabilities across various tasks. However, their extensive parameter count and high inference latency present significant challenges for deployment in resource-constrained edge devices used in driver-assistance systems, automated driving, mobile robotics, or video surveillance. Due to their reduced capacity, smaller models cannot be expected to match the performance of larger ones in arbitrary domains. Additionally, training large-scale foundation models typically involves several millions or billions of images, making it expensive and time-consuming. Together, this motivates the need for smaller domain-specific models, as well as data-efficient training procedures. In this work, we specifically focus on zero-shot image classification, for which only a small-scale image encoder is required. Class-specific text embeddings are fixed and can be precomputed off-device, while only image embeddings are computed on-device. Thus, our goal is to distill smaller drop-in replacements (students) of the CLIP image encoder (teacher) that achieve on-par performance on the specific target domains of interest. In particular, we want to specialize the image encoder student to novel domains for which we only know the relevant classes, but do not have access to actual images, the so called *zero-shot*

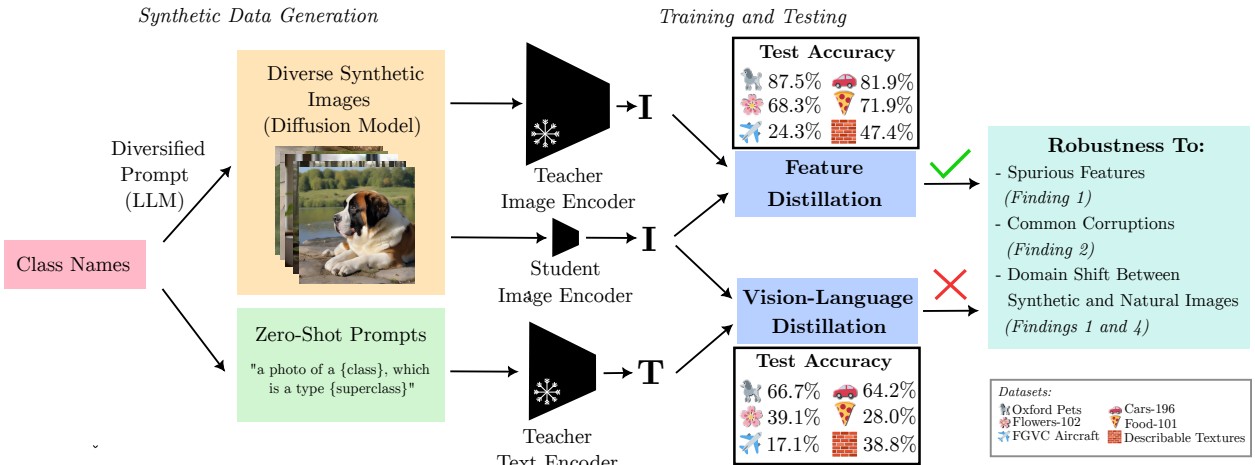

Figure 1: **Overview over our zero-shot distillation framework and the observed properties of vision-language distillation and feature distillation.** We only use the class names to generate domain-specific synthetic images for the distillation of small CLIP image encoders. The reported test accuracies are from domain-specific TinyViT-11M students on six fine-grained classification datasets. The crucial factor to closely reach the performance of the teacher is to minimize the distance between image features of the teacher image encoder and the student encoder (feature distillation). The common approach of aligning the text and image features (vision-language distillation) leads to substantially worse performance. This observation can be attributed to improved robustness properties of feature distillation against spurious features and common corruptions which we discuss in our findings.

*distillation* setting. For zero-shot distillation, domain-specific data can be obtained from "general-purpose" generative models, such as large-scale latent diffusion models (Podell et al., 2023), by class-aware prompting. However, learning from synthetic images has proven challenging (Sariyildiz et al., 2022; Azizi et al., 2023). Despite attempts to improve synthetic images in terms of quality and diversity (Yu et al., 2023) in order to reduce the domain gap to natural images, there still remains a substantial drop in performance when transferring from the synthetic to real image domain (Azizi et al., 2023; Sariyildiz et al., 2022).

While previous works primarily focus on improving the synthetic data for zero-shot learning, we identify the loss function as a critical factor that impacts performance when training on synthetic images. Specifically, we observe that two classes of loss functions exhibit distinct properties when transferring between the synthetic and real domain. Therefore, we differentiate between *vision-language distillation* and *feature distillation*. For vision-language distillation, the embeddings of image-caption pairs are aligned through the loss function. Feature distillation relies solely on matching the image embeddings of a trainable student to those of a frozen teacher. We find that vision-language distillation has several drawbacks, including the susceptibility to spurious features and common corruptions in the training set. This impedes the generalization from a synthetic train to a real test set. Conversely, feature distillation demonstrates greater robustness to these influences. Thereby, it improves the zero-shot transfer from synthetic images. Moreover, we find that the specificity of the data influences the distillation process of image encoders. Current baselines for distilled CLIP image encoders (Wu et al., 2023; Yang et al., 2023a; Vasu et al., 2023) are trained on large-scale common crawl datasets, relying on the models' generalization for zero-shot performance. For small image encoders with a substantially smaller capacity, however, this cannot be achieved. Thus, our approach first distills on *domain-agnostic* datasets with diverse images, such as DataComp (Gadre et al., 2023), ImageNet (Deng et al., 2009) or SynthCI (Hammoud et al., 2024), and subsequently on *domain-specific* synthetic datasets, like pets, food or similarly. This strategy results in superior domain-specific performance, especially for smaller image encoders.

**Contributions and findings.** In this work, we show that small replacements of the CLIP vision encoder can be efficiently and robustly trained using feature distillation on synthetic images. Therefore, we introduce a unifying framework for training vision encoders in a zero-shot setting. The main results are stated in Figure 1 and our key findings within this framework are summarized as follows:

1. **Feature Distillation is Less Susceptible to Spurious Visual Features Than Vision-Language Distillation.** We hypothesize that vision-language distillation is misguided by spurious correlations on the class level. We find evidence for this claim by observing a drop in performance when training on real and synthetic datasets with deliberately introduced spurious features. Unlike vision-language distillation, feature distillation using loss functions such as the $\mathcal{L}_2$ distance between the student and teacher embeddings is robust to these spurious correlations.

2. **Feature Distillation Increases Robustness to Common Corruptions.** We investigate the influence of corruptions on the distilled image encoders. In this respect, feature distillation achieves substantially better robustness compared to vision-language distillation, especially when distilling on synthetic images.

3. **Initial Domain-Agnostic Distillation Accelerates Domain-Specific Distillation.** We find that the process of distilling models on domain-specific synthetic data can be accelerated if the students are distilled on domain-agnostic datasets beforehand. The difference in final performance between using the images of ImageNet (Deng et al., 2009), SynthCI (Hammoud et al., 2024) or DataComp (Gadre et al., 2023) for the domain-agnostic distillation is negligible.

4. **Feature Distillation Bridges the Gap to Baselines Trained on Real Images.** Based on our first three findings, we distill a ViT-B/32 CLIP vision encoder into students based on the TinyViT (Wu et al., 2022) and EfficientNet Tan & Le (2019) architectures with up to 92% fewer parameters using feature distillation on synthetic images. The resulting students closely match the classification performance of the teacher on the Oxford Pets (Parkhi et al., 2012), Flowers-102 (Nilsback & Zisserman, 2008), Stanford Cars (Krause et al., 2013), Food-101 (Bossard et al., 2014), Describable Textures (Cimpoi et al., 2014) and Aircrafts (Maji et al., 2013) datasets. Notably, our students are on par or even surpass the current baselines for distilled CLIP models, including the TinyCLIP model with 8 times more trainable parameters and MobileCLIP which was trained on over 100 times more images using stronger teachers.

5. **Smaller Students Benefit More from Domain-Specific Distillation.** Due to their lower capacity, we observe that small models effectively benefit more from domain-specific distillation in comparison to pure domain-agnostic distillation.

## 2   Related Work

**Knowledge Distillation of Vision-Language Models.**   Knowledge Distillation (Hinton et al., 2015) is a widely used technique for transferring knowledge from larger teachers to smaller students. In its vanilla form, the approach involves combining a standard training loss with a distillation loss that considers the output of both the student and teacher on logit-level, penalizing discrepancies between the two models. Knowledge distillation has been observed to not only benefit the test accuracy of the student on the target datasets but transfer other favorable properties of the teacher such as domain generalization (Ojha et al., 2023). While this approach has been well-established for single-modality tasks including vision (Wu et al., 2022; Mirzadeh et al., 2020; Marrie et al., 2024) or language (Sanh et al., 2020; Jiao et al., 2020), recent works have extended the concept to the multi-modal setting, specifically in the context of vision-language models. CLIP-KD (Yang et al., 2023a) provides an extensive set of experiments comparing various different loss combinations. TinyCLIP (Wu et al., 2023) proposed an advanced initialization process using weight inheritance from the teacher to the student as well as a multi-stage progressive distillation culminating in students that are only one fourth the size of a ViT-B/32 CLIP model. MobileCLIP (Vasu et al., 2023) further refined the distillation process by incorporating image augmentation, synthetic captions, and dedicated architectural choices. In contrast to these existing methods, our approach focuses on finding only a one-to-one replacement of the vision encoder while the text encoder remains frozen. Apart from CLIP-specific techniques, unsupervised distillation based purely on images without labels has been identified as a data-efficient alternative to supervised training for vision encoders (He et al., 2022). We build on this observation

by combining domain-agnostic with domain-specific distillation, both in an unsupervised setting without any labels. Despite knowledge distillation being a widely adopted training technique, it has been observed that it does not always work as commonly understood. Even when the student features the same capacity as the teacher, there can be significant discrepancies in their predictive distributions (Stanton et al., 2021; Ojha et al., 2023).

**Properties of Contrastive Losses.** Several works have highlighted shortcomings of contrastive training. In particular, for contrastive vision-language loss functions Chen & Li (2020) observed the phenomenon of feature suppression where a few easy-to-learn shared features can suppress or prevent the learning of competing features. In a multi-caption setup, Bleeker et al. (2024) highlight that contrastive losses are prone to learning shortcuts contained in the captions. We extend on these observations, by finding that models trained through vision-language distillation are misguided by class-level spurious features and overfit to the synthetic data domain. This impairs the zero-shot generalization properties. Existing approaches that aim at mitigating spurious correlations in multi-modal learning (Yang et al., 2023b) require detecting the spurious features which is inherently difficult in the case of visually not apparent spurious features or corruptions in synthetic images. Our approach using feature distillation increases the robustness to spurious features without having to detect them.

**Training and Distillation Using Synthetic Images.** Recent advancements in generative text-to-image models have sparked a growing interest in the use of synthetic images for vision applications. Azizi et al. (2023) demonstrated that images from fine-tuned text-to-image models can be combined with real images to enhance the accuracy of classifiers on ImageNet-1k (Deng et al., 2009). For text-to-image generation, diffusion models are commonly employed, particularly for knowledge distillation (Li et al., 2023b). However, it was observed that the performance deteriorates, in particular when the number of synthetic images surpasses that of real images or when training only on generated images (Hennicke et al., 2024). Yu et al. (2023) attributed this decline to the lack of diversity in the used synthetic images. To mitigate this issue, they proposed a diversification strategy for the image generation process by incorporating prompts generated by large language models, thereby enhancing content and style variation. Another approach to diversification in the few-shot setting was presented by da Costa et al. (2023), which involved augmentations and low-rank adaptation. By scaling up synthetic datasets, Tian et al. (2023a) and Hammoud et al. (2024) demonstrated the feasibility of training vision-language foundation models solely using images from text-to-image models. However, achieving performance on par with or surpassing models trained on real data necessitates the utilization of a large number of synthetic images, in the order of $10^7$ or $10^8$. This prolongs the already long training process and introduces additional computational overhead. Furthermore, Geng et al. (2024) observe that images for diffusion models frequently posses visual artifacts that deteriorate the performance in comparison to training on images retrieved from common crawl datasets. Importantly, the reported results for models trained on synthetic images are typically obtained by linear probing or after few-shot training and not obtained in zero-shot setting.

## 3 Zero-Shot Distillation for Transfer Learning from Synthetic Images

*Zero-shot distillation* is a framework for transferring knowledge from a teacher to a student in a setting where one does not have access to images from the target domain but only textual descriptions of classes. The framework specifically focuses on the ability of foundation models as teachers to perform well on unseen data due to their generalization properties. The objective is to transfer this performance to a smaller student without utilizing any data from the unseen target domain. Therefore, the primary goal is not to address the disparity between the datasets used to train the teacher and student, but rather to *extract domain-specific knowledge from the teacher without having access to data from the target domain.* The term zero-shot distillation has been introduced previously (Nayak et al., 2019), yet only in the setting for single-modal classifiers that were trained using the cross-entropy loss. In our case, we consider CLIP which is a vision-language model instead of a simple image classifier. For zero-shot distillation, we use two types of teacher knowledge. On the one hand, explicit distillation from a CLIP teacher through the loss function. On the other hand, implicit distillation from the generative text-to-image model that generates the synthetic images. In this section, we discuss the four aspects that constitute the zero-shot distillation framework: the data domain, the training pipeline, the generation of diversified synthetic training data and the selection of an appropriate loss function.

### 3.1 Data Domain

For training in a zero-shot setting, there are currently two core approaches. The first one involves relying on large-scale, domain-agnostic data such as common crawl datasets (Schuhmann et al., 2021; Gadre et al., 2023). While this approach is feasible for large foundation models, it poses challenges for smaller models as these lack the capacity to fit diverse data to the same extent as larger ones. The second approach involves domain-specific distillation using either purely synthetic images (Hammoud et al., 2024; Tian et al., 2023b;a) or a combination of real and synthetic images (Yu et al., 2023; Azizi et al., 2023). Yet, the existing works that use this approach incorporate few-shot learning on real images (Hammoud et al., 2024; Tian et al., 2023b) or linear probing (Hammoud et al., 2024; Tian et al., 2023a;b) after training on synthetic data. Consequently, the reported accuracies are no longer zero-shot. Within our framework, we find that best zero-shot performance is achieved through a two-stage approach: first training on domain-agnostic images, followed by training on domain-specific synthetic images. An alternative to using synthetic images would be to retrieve images from the dataset on which the generative model was trained on. However, this dataset might be proprietary or strictly regulated due to privacy concerns such that synthetic images are the only feasible option. Thus, we focus on using synthetic images for domain-specific distillation in our framework.

### 3.2 Training Pipeline

In order to shorten training in comparison to training from scratch, Wu et al. (2023) introduced weight inheritance as an initialization scheme for distilling CLIP models. This method has a significant limitation as it can only be applied when the student shares a similar architecture with the teacher. Instead of using weight inheritance, our framework comprises of training on a domain-agnostic dataset, which is also referred to as pre-training (Gan et al., 2022), before training on domain-specific synthetic datasets. Training large foundation models like the original CLIP (Radford et al., 2021) typically requires substantial computational resources due to the use of billions of images. Yet, He et al. (2022) observed that domain-agnostic distillation on natural images can be sped up significantly by using feature distillation. For our purpose of distilling CLIP vision encoders using synthetic images, this step has further advantages: by aligning the teacher and student image features, we can mitigate phenomena like the modality gap (Liang et al., 2022) where corresponding output vectors are located in different areas of the embedding space. Through feature distillation, we ensure that the students geometrically match the teacher and can be used as direct replacements. After domain-agnostic distillation, we perform the domain-specific distillation on synthetic images. Domain-agnostic distillation only needs to carried out once for all students. The additional costs compared to distillation from scratch only on the synthetic images are therefore negligible if domain-specific distillation is performed for many target domains.

### 3.3 Data Diversification of Synthetic Images

In the context of zero-shot learning for image classification, synthetic data generation is based on the class names. However, it has been observed that using only the names to generate images using diffusion models leads to suboptimal performance (Sariyildiz et al., 2022). This is primarily due to the lack of diversity in the generated images as well as class ambiguity (da Costa et al., 2023). To address this challenge, recent approaches have turned to leveraging large language models (LLMs) to enhance diversity in the prompts. In addition to class names, LLMs are guided by additional inputs for diversification, such as information from a concept bank (Hammoud et al., 2024) or specific requirements related to contextual and style diversification (Yu et al., 2023). By incorporating these additional sources of guidance, the generated synthetic data becomes more diverse and aligned with the desired objectives of the target setting. Using the approach from Yu et al. (2023), our framework focuses on *contextual dimensions* to achieve diversification. These dimensions are attributes that describe the context of the image such as the background, camera angle, object position, presentation style, and superclasses, all of which are tuned specifically for the target dataset. In contrast to Yu et al. (2023), we do not prompt the LLM for each caption separately, but ask for different options for each contextual dimension. This reduces the risk of obtaining similar captions. The final prompt used for the text-to-image model is a comma-separated list of options for these contextual dimensions. Instead of using all possible combinations of options, which would result in a strongly growing number of images given more options, we use an approach based on combinatorial testing (Ahmed et al., 2017; Nie & Leung, 2011) described in Section A.3.

| Name | Evaluation Metric | Domain-Agnostic | Domain-Specific | Natural Images | Synthetic Images | Data Diversification | Loss |
|---|---|---|---|---|---|---|---|
| StableRep (Tian et al., 2023b) | Linear probe, few-shot | ✓ | | | ✓ | | MP |
| SynCLR (Tian et al., 2023a) | Linear probe | ✓ | | | ✓ | ✓ | MP |
| SynthCLIP (Hammoud et al., 2024) | Linear probe, few-shot | ✓ | | | ✓ | ✓ | CLIP |
| Fake it till you make it (Sariyildiz et al., 2022) | Zero-Shot Acc. | ✓ | | | ✓ | | Cross-entropy |
| Diversify don't finetune (Yu et al., 2023) | Accuracy | ✓ | | ✓ | ✓ | ✓ | Custom |
| (Azizi et al., 2023) | Accuracy | ✓ | | ✓ | ✓ | | Cross-entropy |
| DM-KD (Li et al., 2023b) | Accuracy* | | ✓ | | ✓ | | Logit-based knowledge distillation |
| TinyCLIP (Wu et al., 2023) | Zero-Shot Acc. | ✓ | | ✓ | | | CLIP, Affinity mimicking |
| MobileCLIP (Vasu et al., 2023) | Zero-Shot Acc. | ✓ | | ✓ | | | CLIP, Affinity mimicking |
| **Zero-Shot Distillation** (Ours) | Zero-Shot Acc. | ✓ | ✓ | ✓ | ✓ | ✓ | $\mathcal{L}_2$ feature distillation |

Table 1: **Our framework differs from previous approaches by using feature distillation instead of vision-language distillation.** For DM-KD, the teacher was trained on the real domain-specific images, which is not possible in a zero-shot setting. This is symbolized by *.

## 3.4 Loss Selection

The choice of loss function distinguishes our framework from existing approaches for distillation in the zero-shot setting. The critical distinction lies between vision-language distillation and feature distillation. The latter ultimately improves effective transfer learning from synthetic images which we demonstrate through our five findings in Section A.9. Therefore, our framework is based on feature distillation.

**Vision-Language Distillation.** Training image encoders is typically carried out using loss functions such as the cross-entropy loss that consider image-class pairs and tries to optimize for correct class predictions. The training of small models can be improved by adding guidance from a larger model through knowledge distillation. Standard knowledge distillation compares the predictive distribution between student and teacher over the classes. When training image encoders or vision-language models on common crawl, class-based loss functions are not applicable as the images are paired with captions instead of class labels. For this purpose, the contrastive InfoNCE loss (van den Oord et al., 2019), also know as CLIP loss, aims at aligning the embeddings between image and caption. In the case of training on datasets with image-class pairs, the CLIP loss can still be applied by using the zero-shot captions "a photo of {class name} which is a type of {superclass}". These were originally introduced for the zero-shot inference of the original CLIP model (Radford et al., 2021). "Superclass" refers to a general description of the object that can be encountered such as pets, food, cars or similarly. By using these class-specific prompts, several images in a batch may share the same caption. This conflicts the goal of decreasing the similarity of image embeddings to the text embeddings of not matching captions in the CLIP loss. An alternative to the CLIP loss is given by the multi-positive contrastive loss introduced in StableRep (Tian et al., 2023b). The details on how to adapt the multi-positive (MP) loss to our setting are given in Section A.15. In the following, we refer to the CLIP and MP losses as vision-language losses.

**Feature Distillation.** (Romero et al., 2014) found that the generalization and training speed of thin convolutional neural networks can be improved by adding an additional loss that aligns the features of the student and teacher. In contrast to the vision-language distillation, the student directly learns from the image features of the teacher without considering the captions. Like He et al. (2022), our framework builds on the $\mathcal{L}_2$ distance between the normalized student and teacher image features as loss function. This choice is motivated by the theoretical investigation presented in Section A.16, which highlights that low $\mathcal{L}_2$ loss and high train-test similarity are sufficient to ensure student-teacher agreement. An alternative loss function for feature distillation is investigated in Section A.7.

## 3.5 Positioning of Existing Approaches in our Framework

To complement the framework, we position existing baselines with respect to the discussed components in Table 1. The first difference between existing works and our setup is that apart from DM-KD (Li et al., 2023b), none of the other approaches perform distillation on domain-specific datasets. The most crucial aspect is that all baselines using synthetic images are trained with vision-language losses whereas we use feature distillation. Based on our findings presented in Section 5, these differences are decisive for enabling effective transfer learning from synthetic images.

## 4 Experimental Setup

In this section, we briefly describe the setup used to conduct the experiments supporting our findings.

**Datasets and Hyperparameters.** As introduced in Section 3.2, the first step of our framework is to perform feature distillation on a large-scale, domain-agnostic dataset. For this purpose, we select DataComp medium (Gadre et al., 2023) with 123 million images and train for a single epoch. At the time we conducted our experiments 86% of the original image URLs were still active. For comparison, we perform to domain-agnostic distillation on ImageNet (Deng et al., 2009) with 1.28 million images and SynthCI 30M Hammoud et al. (2024) with 30 million synthetic images in Section 5.3. For domain-specific distillation, we target the Oxford Pets (Parkhi et al., 2012), Oxford Flowers (Nilsback & Zisserman, 2008), Food-102 (Bossard et al., 2014), Stanford Cars (Krause et al., 2013), Describable Textures (Cimpoi et al., 2014) and Aircrafts (Maji et al., 2013) datasets. In the appendix, we include ImageNet-100 (Tian et al., 2020) as a non-domain-specific dataset for reference. These datasets are only used for testing while the actual datasets used for training are synthetically generated based on the class names. Using the diversification strategy discussed in Section 3.3, we select a set of five different contextual dimensions and corresponding weights in the prompts for the diffusion model. The number of images per class roughly matches the size of the real training datasets. We use 265 images per class for the smaller, less diverse datasets and 1011 for the larger ones. More details on the selection of contextual dimension and the dataset sizes are given in Section A.3. As the selection of options for the contextual dimensions and superclasses are relatively simple, we can use a smaller language model Llama-2 7B fine-tuned for chats (Touvron et al., 2023) and still obtain sufficiently diverse prompts. For the generation of the images, we utilize a LCM LoRA (Luo et al., 2023) of Stable Diffusion XL (Podell et al., 2023). Due to the LCM LoRA architecture, 6 inference steps together with a guidance scale of 0.5 suffice to obtain high-quality images. For both domain-agnostic and domain-specific distillation we use the same hyperparameters. We train using a batch size of 256 and a constant learning rate of $5 \times 10^{-4}$ using the AdamW optimizer (Loshchilov & Hutter, 2019). All other hyperparameters and augmentations were kept consistent with the CLIP training methodology (Radford et al., 2021). One epoch of training on DataComp medium corresponds to $4.3 \times 10^5$ optimization steps. For domain-specific distillation, we perform 96 optimization epochs for all models. Ablations with a different teacher model and synthetic data generator are contained in the supplementary material.

**Student and Teacher Architectures.** As teacher, we employ a ViT-B/32 (Dosovitskiy et al., 2021) CLIP vision encoder that was trained on DataComp-XL, a dataset consisting of 12.8 billion image-text pairs from common crawl (Gadre et al., 2023). The corresponding text encoder follows the same architecture as described in the original CLIP paper, with 63 million parameters (Radford et al., 2021) and an embedding dimension of 512. For our students, we utilize two different types of state-of-the-art architectures: Efficient-Nets (Tan & Le, 2019), which are based on convolutional neural networks, and TinyViTs (Tan & Le, 2019), which are hybrid models combining convolutions and transformers. For our final results, we respectively select three models in the 5, 10, and 20 million parameter range from each architecture type. To present our findings, we report the results on the TinyViT with 11 million parameters. To align the output of the vision encoder with the embedding dimension of the teacher, we apply a single linear projection head.

## 5 Findings

In this section, we present our five main findings that provide insights into why and how feature distillation greatly improves transfer learning from synthetic images.

### 5.1 *Finding 1:* Feature Distillation is Less Susceptible to Spurious Visual Features Than Vision-Language Distillation

Our first set of experiments is designed to test the hypothesis that class-level information introduced through the captions in vision-language distillation leads the model to learn spurious features as well as characteristics of synthetic images. To validate our hypothesis, we conducted two experiments on the pets dataset, deliberately introducing dedicated spurious features into real and synthetic images.

| Train Data | Test Data | feature distillation $\mathcal{L}_2$ | vision-language distillation CLIP | MP | feature and vision-language distillation combined $\mathcal{L}_2$+CLIP | $\mathcal{L}_2$+MP | Teacher |
|---|---|---|---|---|---|---|---|
| Real *with* spurious features | Real *without* spurious features | 88.9 | 60.0 | 77.3 | 88.0 | **90.0** | 89.8 |
| | Real *with* spurious features | 90.3 | 96.5 | 96.5 | 95.7 | 88.3 | 89.6 |
| | Real *with* shuffled spurious features | 88.0 | 48.7 | 51.7 | 88.0 | **88.6** | 88.7 |
| Synth. *with* spurious features | Real *without* spurious features | **84.4** | 24.3 | 31.0 | 81.2 | 35.6 | 89.8 |
| | Synthetic *with* spurious features | 94.2 | 100.0 | 100.0 | 99.3 | 100.0 | 93.3 |
| | Synthetic *without* spurious features | 90.9 | 53.6 | 61.7 | **91.4** | 66.7 | 93.9 |

Table 2: **Feature distillation increases the robustness to spurious features.** The accuracies refer to students distilled and evaluated on pets. The spurious features are introduced through adding colored shapes (real) or class-specific unicolor backgrounds (synthetic). For the test set "shuffled spurious features" the coloured shapes are shuffled among the classes. Red indicates a performance drop due to the reliance on spurious features.

**Natural Images with Spurious Features.** To investigate the impact of spurious features in the natural image domain, we add class-specific colored shapes to the images in the pets dataset. These shapes were added to each image in the training split. Examples can be seen in Figure 2. Using these images, we perform domain-specific distillation with different losses after the domain-agnostic distillation. The test accuracies of the resulting models are shown in Table 2. We observe a decrease in performance on the test set without spurious features when the students were trained with vision-language distillation using the CLIP or multi-positive (MP) loss. This suggests that these students did not acquire any additional class-specific features during domain-specific distillation with vision-language losses. Instead, they overfit on the visual spurious features. When evaluating these students on a test set of real images where the colored shapes are shuffled between classes, we observe a significant degradation in performance. In contrast, the students trained with the $\mathcal{L}_2$ loss achieves accuracies comparable to the dataset without spurious features on both test sets. These findings highlight the robustness of feature distillation in mitigating the negative impact of spurious features in the natural image domain.

**Synthetic Images with Spurious Features.** To investigate whether the observed behavior on real images can be replicated with synthetic ones, we generated a synthetic dataset incorporating dedicated spurious features. Specifically, we sample images where pets are positioned against a solid-colored background, with each class assigned a distinct color. The results shown in Table 2 indicate that the performance of students trained with vision-language distillation deteriorates when confronted with the presence of these spurious features. Figure 2 showcases instances of misclassifications. Importantly, the student trained through feature distillation exhibited a test accuracy of 84.4%, which is only 5% lower than the accuracy of the teacher despite the domain gap between real and synthetic images, as well as the presence of spurious features.

## 5.2 *Finding 2:* Feature Distillation Increases Robustness to Common Corruptions

In addition to the influence of spurious features, we hypothesize that feature distillation increases robustness against common corruptions. In order to evaluate this claim, we conducted a comprehensive benchmark study and assessed the performance of the classifiers under 15 common corruptions and five severity levels (Hendrycks & Dietterich, 2019) on the pets dataset. In Table 3 we report the relative performance under corruption with respect to the classification accuracy as defined by (Michaelis et al., 2019). Further results are provided in the Supplementary Section A.9. Our observations reveal that feature distillation improves the robustness, regardless of whether training is performed on real or synthetic data. The distinction to the models trained with vision-language losses is particularly prominent when training on domain-specific synthetic data. In this case, the models trained with the CLIP loss perform worse than the models trained purely on domain-agnostic data. The models trained using only feature distillation achieve the strongest robustness which reflects the observations from Sections 5.1.

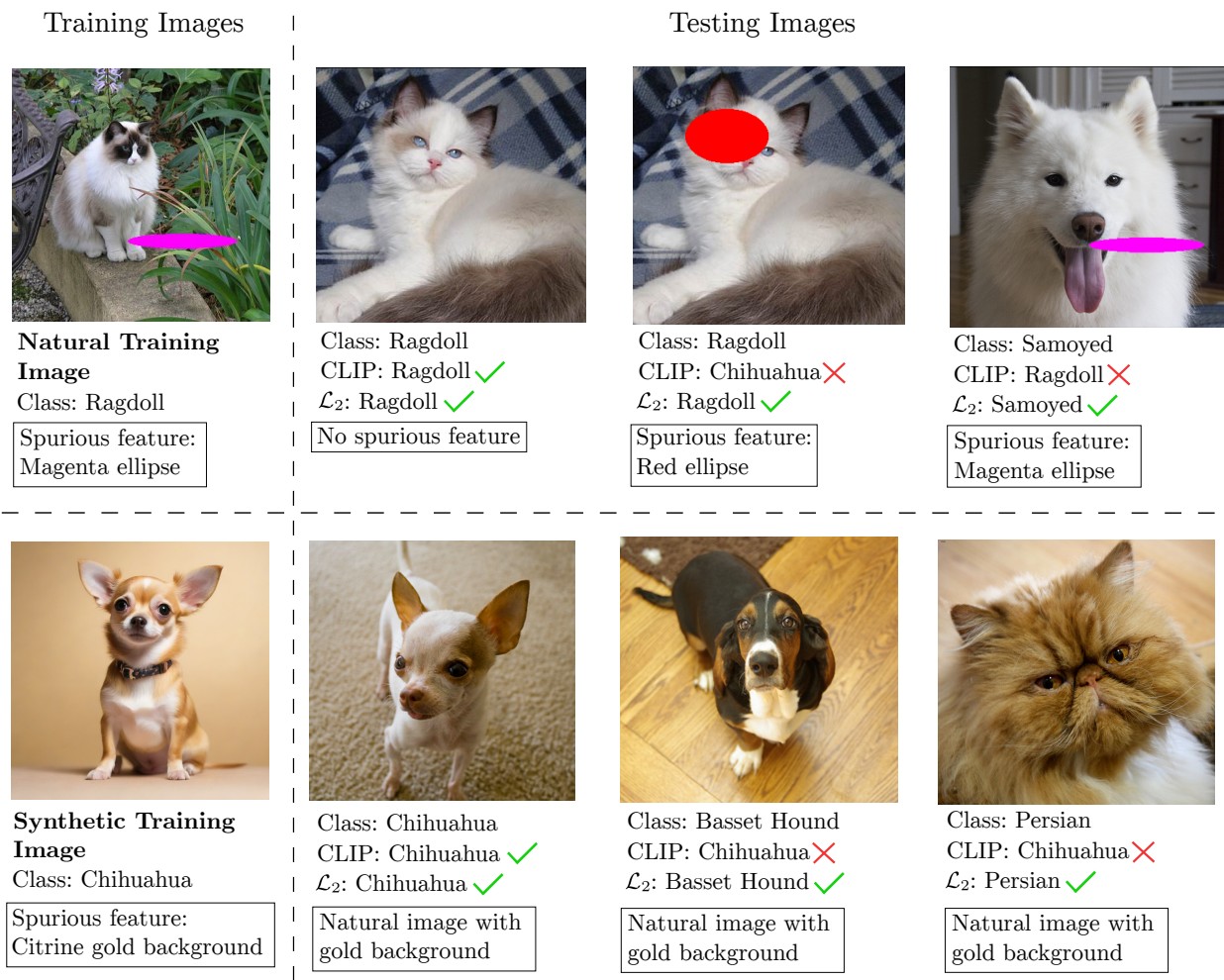

Figure 2: **Examples for the influence of spurious features in the training images.** Examples for the correct and incorrect classifications. The first row corresponds to the setting of training on natural images with colored shapes spurious features. The second corresponds to students trained on synthetic images where the background colors are spurious features. All of the test examples are classified correctly by the teacher and the students trained through $\mathcal{L}_2$ feature distillation.

| Domain-Specific Train Data | Test Data | feature distillation $\mathcal{L}_2$ | vision-language distillation CLIP | MP | feature and vision-language distillation combined $\mathcal{L}_2$+CLIP | $\mathcal{L}_2$+MP | Teacher |
|---|---|---|---|---|---|---|---|
| Real | Real (corrupted) | 0.82 | 0.78 | 0.79 | **0.86** | 0.79 | 0.85 |
| Synthetic | Real (corrupted) | **0.79** | 0.49 | 0.50 | 0.77 | 0.52 | - |

Table 3: **Feature distillation increases robustness to common corruptions.** Relative performance under corruption with respect to the classification accuracy on the pets dataset under 15 common corruptions and five severity levels as defined by (Michaelis et al., 2019)

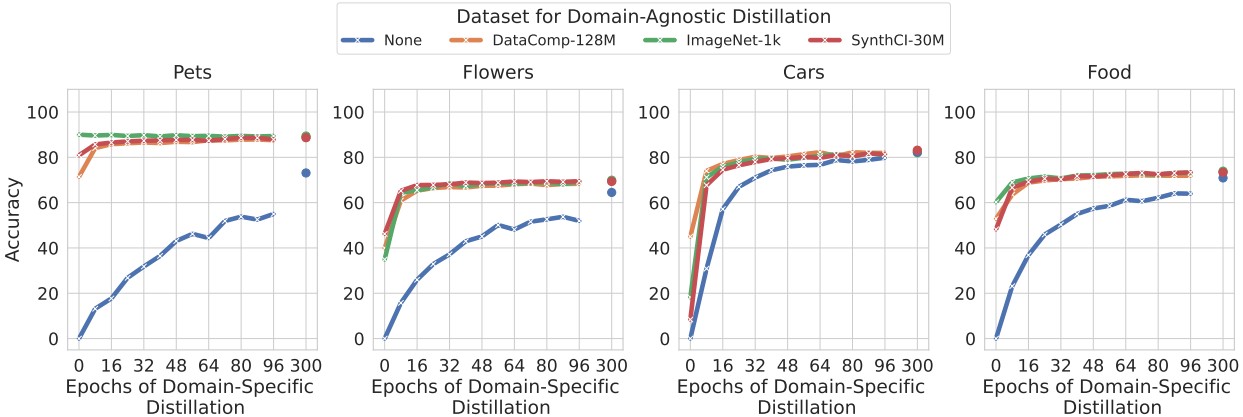

Figure 3: **Initial domain-agnostic distillation on large-scale datasets of natural images accelerates subsequent domain-specific distillation on synthetic images.** We perform domain-specific distillation of TinyViT 11M students on synthetic images and report the zero-shot classification accuracy depending on the number of epochs. The students are either distilled from scratch (no domain-agnostic distillation) or after initial domain-agnostic distillation on ImageNet, DataComp medium or SynthCI 30M.

---

### 5.3 *Finding 3:* Initial Domain-Agnostic Distillation Accelerates Domain-Specific Distillation

To complement our previous findings, we investigate the influence of domain-agnostic distillation on the domain-specific performance of the students. Therefore, we compare domain-agnostic distillation on DataComp medium, ImageNet and SynthCI 30M as well as no domain-agnostic distillation (domain-specific distillation from scratch). In Figure 3, the accuracies depending on the number of epochs are shown for students trained on domain-specific synthetic data. First, we find that when distilling for less than 100 epochs, initial domain-agnostic distillation consistently leads to higher accuracies. Furthermore, the accuracy of the models with initial domain-agnostic distillation converges within fewer epochs. When performing domain-specific distillation for 300 epochs, the models without domain-agnostic distillation come closer to the performance of the models with domain-agnostic distillation. The difference on the cars and food datasets is within 3%. On the pets and flowers dataset, however, the margin still remains greater than 5%. Domain-agnostic distillation only needs to be performed once for all students. Therefore, its additional costs become negligible if domain-specific distillation is performed for many target domains. Second, we find that the data with which the domain-agnostic distillation is performed has a marginal impact on the performance of the models after domain-specific distillation. The difference of the final performance (after domain-specific distillation) using domain agnostic-distillation on ImageNet or SynthCI versus DataComp is smaller than 2% on all four datasets. In contrast, the accuracy of the students distilled only on the domain-agnostic datasets differs more strongly. For example, the model distilled on ImageNet is 19% better on pets but 21.7% worse on cars compared to the model that was distilled on DataComp medium. This can be explained by the overlap between classes in the train and test datasets. 25 of the 37 classes from Oxford Pets are already contained in ImageNet. This also explains why domain-specific distillation on pets after distillation on ImagNet leads to a slight decrease in performance. In contrast, none of the classes of the cars dataset are part of ImageNet. The student distilled on the synthetic images from SynthCI 30M achieves accuracies similar to the student distilled on DataComp medium apart from the cars dataset where the SynthCI model only reaches 8.4% accuracy. Regardless of whether the initial domain-specific distillation was performed on synthetic images or real images, the difference in performance of the students after subsequent domain-specific distillation is marginal. This indicates that the decisive factor for accelerating the domain-specific distillation is the diversity of the images for the domain-agnostic distillation and not whether they are real or synthetic.

| | Model | Loss | Training Dataset | Trainable ImgEnc | Params. TxtEnc | #Samples Seen | Pets | Flowers | Cars | Food |
|---|---|---|---|---|---|---|---|---|---|---|
| CLIP | ViT-B/32 | CLIP | DataComp-XL | 86M | 63M | 12.8B | 89.7 | 72.9 | 85.4 | 82.9 |
| | ViT-B/32 | CLIP | DataComp-medium | 86M | 63M | 128M | 43.1 | 29.7 | 28.0 | 41.7 |
| | RN-50 | CLIP | openai | 86M | 63M | 32×400M | 85.3 | 65.2 | 54.5 | 80.8 |
| TinyCLIP | ViT-61M/32-29M | CLIP+AM | LAION-400M | 61M | 29M | 38×400M | 87.3 | 64.7 | 79.1 | 73.4 |
| | ViT-40M/32-19M | CLIP+AM | LAION-400M | 40M | 19M | 38×400M | 84.4 | 61.0 | 74.2 | 71.4 |
| | ViT-8M/16-3M | CLIP+AM | YFCC-15M | 8M | 3M | 50×15M | 45.8 | 57.4 | 8.0 | 56.2 |
| | RN-19M-19M | CLIP+AM | LAION-400M | 19M | 19M | 12×400M | 81.0 | 56.4 | 70.1 | 66.7 |
| Mobile CLIP | MobileCLIP-S2 | CLIP+AM | DataCompDR | 56M | 63M | 13B | 92.7 | 74.7 | 86.2 | 86.8 |
| | MobileCLIP-S1 | CLIP+AM | DataCompDR | 22M | 63M | 13B | 93.1 | 72.9 | 84.2 | 84.9 |
| | MobileCLIP-S0 | CLIP+AM | DataCompDR | 11M | 42M | 13B | 89.9 | 67.8 | 79.4 | 79.1 |
| Domain-Agnostic | TinyViT-11M | CLIP | DataComp-medium | 11M | - | 110M | 10.4 | 4.2 | 5.4 | 4.7 |
| | TinyViT-11M | $\mathcal{L}_2$ | DataComp-medium | 11M | - | 110M | 71.4 | 39.9 | 45.0 | 52.9 |
| | TinyViT-11M | $\mathcal{L}_2$ | DataComp-medium | 11M | - | 5× 110M | 78.4 | 50.0 | 58.7 | 61.1 |
| Domain-Agnostic + Domain-Specific | TinyViT-11M | $\mathcal{L}_2$ / CLIP | DataComp-medium + *Synthetic* Training Images | 11M | - | 110M +1M-9M | 66.7 | 39.1 | 64.2 | 28.0 |
| | TinyViT-11M | $\mathcal{L}_2$ / $\mathcal{L}_2$ | DataComp-medium + *Synthetic* Training Images | 11M | - | 110M +1M-9M | 87.5 | 68.3 | 81.9 | 71.9 |
| | TinyViT-11M* | $\mathcal{L}_2$ / CLIP | DataComp-medium + *Real* Training Images | 11M | - | 110M +1M-7M | 88.0 | 90.6 | 90.7 | 89.1 |
| | TinyViT-11M* | $\mathcal{L}_2$ / $\mathcal{L}_2$ | DataComp-medium + *Real* Training Images | 11M | - | 110M +1M-7M | 88.7 | 68.4 | 83.8 | 83.0 |

Table 4: **Our models trained on synthetic images bridge the gap to training on natural images.** The upper part summarizes the baseline CLIP, TinyCLIP and MobileCLIP. "Domain-agnostic" denotes distillation only on DataComp medium. "Domain-agnostic+Domain-specific" contains the results where either synthetic data or real data is used for subsequent domain-specific distillation. The blue box highlights our final models using synthetic images. Gray numbers indicate that the performance is not zero-shot.

## 5.4  *Finding 4:* Zero-Shot Distillation Bridges the Gap to Baselines Trained on Real Images

Our goal is to achieve state-of-the-art zero-shot accuracy on fine-grained visual classification tasks. Based on findings 1 and 2, we hypothesize that feature distillation greatly improves zero-shot training of small vision encoders on synthetic data. Therefore, we report the zero-shot classification accuracy of image encoders based on TinyViT-11M architecture and compare them to existing baselines. The results are shown in Table 4 with additional datasets presented in Table 5.

**Baselines.**  The main baseline is the performance of the ViT-B/32 teacher trained on DataComp-XL and the same model trained on DataComp-medium. The teacher achieves zero-shot accuracies of over 80% on the pets, cars and food datasets as well as over 70% on the flowers dataset. The accuracies of the ViT/B-32 CLIP model trained on DataComp medium are substantially worse. The performance gap is between 28% and 42%. Additionally, we report the accuracies of four TinyCLIP and three MobileCLIP models. These models have undergone extensive training on large-scale datasets for multiple epochs. In the case of TinyCLIP, the LAION-400M (Schuhmann et al., 2021) or YFCC-15M (Thomee et al., 2016) datasets were used. Even the smallest TinyCLIP model has been exposed to six times as many images as our models, while the largest models have encountered over 120 times as many samples. The TinyCLIP models exhibit a comparable size to our models in terms of the number of parameters when not considering the text encoder which is not required for zero-shot classification. The largest TinyCLIP model has 40% fewer parameters than the ViT-B/32 CLIP model and achieves its performance up to a margin of 9%. The smallest TinyCLIP model features the same number of trainable parameters as our students but has a gap of over 75% to the ViT-B/32 CLIP model on the cars dataset. Of the MobileCLIP models, the smallest one features an image encoder that is comparable in size to our students. Its performance, however, is comparable to the largest TinyCLIP model. This can be attributed to the fact that it was trained on a dataset featuring the same number of images as DataComp-XL but with synthetically enhanced captions and additional augmentations. Furthermore, the MobileCLIP models are not distilled from a single teacher but from an ensemble of ViT-L teachers, which are stronger than the teacher of our models. At the time of conducting our experiments, we were unable to compare our results with CLIP-KD (Yang et al., 2023a) as these models were not publicly available.

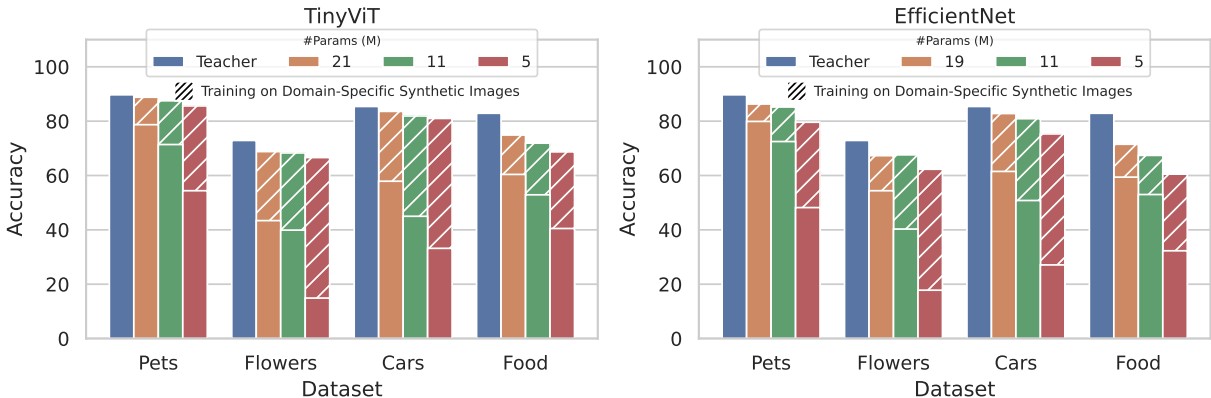

Figure 4: **Domain-specific distillation is more effective for models with fewer parameters.** Zero-shot classification performance of the students after domain-agnostic distillation on DataComp medium for one epoch (solid) and after subsequent domain-specific distillation on the synthetic datasets for 96 epochs (hatched). All experiments were performed using feature distillation.

**Our models.** From our framework, we report two types of models: domain-agnostic distillation and domain-agnostic followed by domain-specific distillation. Based on finding 3, we expect the best performance from the second group. For reference, we include models that were domain-specifically distilled on the complete real datasets as well. However, these accuracies are *not* zero-shot. First, we observe that vision-language distillation using the CLIP loss results in substantially worse performance both in the domain-agnostic and domain-specific case. This reflects our findings 1 and 2 and provides justification to base our framework solely on feature distillation. We find that the resulting models outperform even the largest TinyCLIP model on three of the four datasets despite having 88% fewer trainable parameters. Moreover, they achieve comparable performance to the teacher with a margin of 5% on the same datasets. The larger performance gap on the food dataset can likely be attributed to the more diverse and larger test set. When comparing to ViT-B/32 trained on DataComp-medium, which has been trained on a comparable number of images, even the models that were distilled purely on domain-agnostic data demonstrate substantially superior performance. The MobileCLIP-S0 model features a similarly large image encoder as our models and achieves comparable accuracies despite being trained on a much larger dataset with stronger teachers.

---

**5.5 *Finding 5:* Smaller Students Benefit More from Domain-Specific Distillation**

---

Based on the fact that small image encoders have a lower capacity, we formulate our fifth hypothesis: for models with fewer parameters, domain-specific distillation is more effective when comparing to pure domain-agnostic distillation. To test this, we report the zero-shot performance of five additional students after domain-agnostic and domain-specific feature distillation in Figure 4. There is a general trend of improved performance with increasing model size. Yet, we observe that the effectiveness of domain-specific distillation is more pronounced for smaller models compared to larger ones. The difference in performance between the largest and smallest models is around 10% to 15% after domain-specific distillation, while after only domain-agnostic distillation it is as high as 30%. These findings support our claim that domain-specific distillation is particularly effective for smaller students since it allows them to adapt to the target domain, without requiring real in-domain data.

## 6 Conclusion

In this work, we introduced a framework for distilling small CLIP image encoders in a zero-shot setting using synthetic images. We identify vision-language distillation as a potentially detrimental factor for generalization capabilities of models between synthetic and real data, due to exploitation of spurious features and the susceptibility to common corruptions. By employing feature distillation, we successfully mitigate these limitations. As a result, we are able to train models that surpass the current state-of-the-art for zero-shot CLIP distillation.

**Limitations and Future Work.** In our work, we show that feature distillation is sufficient to greatly improve the transfer performance of small image encoders between synthetic and real data. However, as image generators continue to improve, the synthetic-to-real gap is becoming smaller which could create scope for alternative approaches. To broaden the applicability of our framework, future research could extend zero-shot distillation beyond CLIP image encoders to encompass other computer vision tasks. For instance, architectures such as BLIP-2 (Li et al., 2023a) or LLava (Liu et al., 2024; 2023a;b) are built on top of CLIP image encoders. In these models, however, the image encoder only accounts for a small portion of the overall model size, so that the remaining architectural components must also be considered for model compression. Another potential application of our findings is in the context of dataset distillation where, as in our zero-shot distillation framework, training is performed on synthetic data in which correlations and biases may be present (Cui et al., 2024).

## Acknowledgements

We would like to thank Nicole Finnie for support on pre-training CLIP models as well as Lukas Schott and Maximilian Müller for helpful discussions on the manuscript. We also thank the European Laboratory for Learning and Intelligent Systems (ELLIS) for supporting Niclas Popp.

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

# A  Appendix

## A.1  Additional Datasets

In addition to the results presented in Section 5, we report the accuracies on three additional datasets. One one-hand, on the Describable Textures (Cimpoi et al., 2014) and Aircraft (Maji et al., 2013) dataset, which contain fine-grained, domain-specific classes, as well as ImageNet-100 (Tian et al., 2020), which is a non-domain-specific subset of ImageNet (Deng et al., 2009). The results on textures and aircrafts consolidate our findings from Section 5. On ImageNet-100, domain-specific feature distillation yields almost the same zero-shot accuracy as pure domain-agnostic distillation. This is different to the domain-specific datasets where we could observe a consistent improvement. Presumably, this can be attributed to the greater diversity of the real test images in ImageNet-100, which is not sufficiently captured by the synthetic training data.

## A.2  Evaluation on the Retrieval Task

In addition to image classification, we consider retrieval on MSCOCO (Lin et al., 2014) as a further down-stream task for CLIP models. To evaluate our students, we consider two settings. First, the standard text-to-image and text-to-image retrieval tasks on the entire validation set. The goal of text-to-image retrieval is to select images from a large pool that best match a given caption (the so-called query). For image-to-text retrieval, the setup is the other way around, i.e. querying with an image and selecting relevant captions. With CLIP models, text-to-image retrieval is performed by selecting the images or captions from the pool whose embeddings have the highest cosine similarity to the query. For this, the image encoders need strong domain-agnostic performance, since all images have to be encoded in a meaningful way. The images are *not* restricted to a specific domain. Therefore, we refer to this setting as *domain-agnostic retrieval*. As second setting, we consider a domain-specific image-to-text retrieval task for which only images from a spe-

| | Model | Loss | Training Dataset | DTD | Aircraft | ImageNet-100 |
|---|---|---|---|---|---|---|
| CLIP | ViT-B/32 | CLIP | DataComp-XL | 54.6 | 23.9 | 86.1 |
| | ViT-B/32 | CLIP | DataComp-medium | 20.6 | 3.1 | 49.6 |
| | RN-50 | CLIP | openai | 39.5 | 17.4 | 77.5 |
| TinyCLIP | ViT-61M/32-29M | CLIP+AM | LAION-400M | 49.4 | 17.4 | 81.1 |
| | ViT-40M/32-19M | CLIP+AM | LAION-400M | 49.1 | 13.5 | 79.8 |
| | ViT-8M/16-3M | CLI+AMP | LAION-400M | 26.3 | 7.0 | 65.5 |
| | RN-19M-19M | CLIP+AM | YFCC-15M | 45.3 | 13.2 | 77.3 |
| Mobile CLIP | MobileCLIP-S2 | CLIP+AM | DataCompDR-1B | 30.4 | 59.8 | 90.4 |
| | MobileCLIP-S1 | CLIP+AM | DataCompDR-1B | 27.2 | 59.0 | 89.7 |
| | MobileCLIP-S0 | CLIP+AM | DataCompDR-1B | 53.8 | 20.2 | 85.4 |
| DA | TinyViT-11M | CLIP | DataComp-medium | 19.3 | 1.2 | 22.6 |
| | TinyViT-11M | $\mathcal{L}_2$ | DataComp-medium | 45.1 | 3.4 | 74.3 |
| Domain-Agnostic + Domain-Specific | TinyViT-11M | $\mathcal{L}_2$ CLIP | DataComp-medium + Synthetic | 38.8 | 17.1 | 53.2 |
| | TinyViT-11M | $\mathcal{L}_2$ $\mathcal{L}_2$ | DataComp-medium + Synthetic | 47.4 | 24.3 | 74.0 |
| | TinyViT-11M* | $\mathcal{L}_2$ CLIP | DataComp-medium + *Real* Train Images | 69.7 | 58.6 | 87.0 |
| | TinyViT-11M* | $\mathcal{L}_2$ $\mathcal{L}_2$ | DataComp-medium + *Real* Train Images | 53.0 | 23.6 | 81.8 |

Table 5: **Three additional datasets consolidate our findings.** As in Table 4, the upper part summarizes the baseline CLIP, TinyCLIP and MobileCLIP. "DA" denotes domain-agnostic distillation for one epoch on DataComp medium and "domain-specific" contains the results where either synthetic data (zero-shot) or real data is used for domain-specific distillation of the model. The blue box highlights our final models. Gray numbers indicate that the performance is not zero-shot.

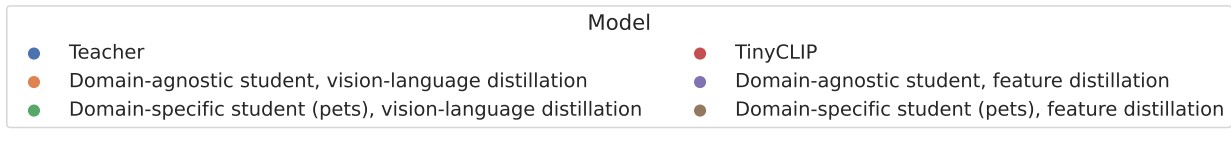

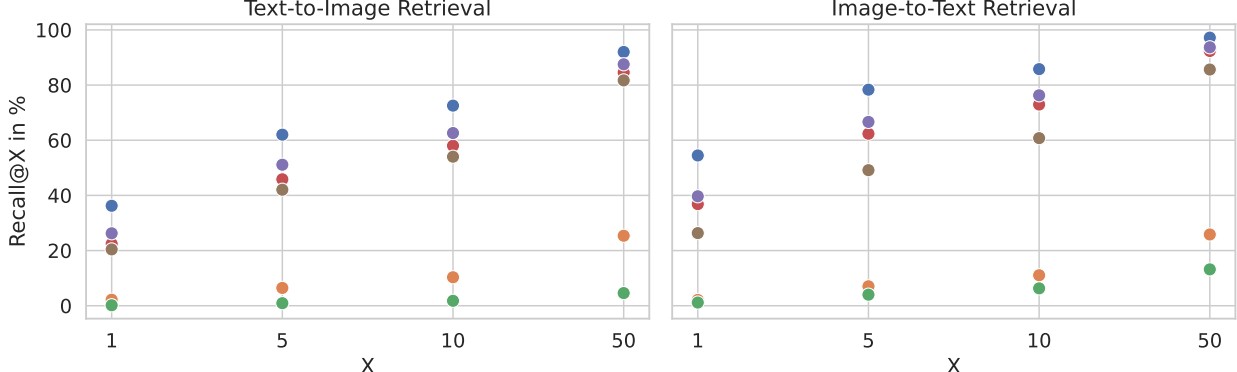

Figure 5: **Feature distillation improves domain-agnostic retrieval on MSCOCO.** We evaluate four of our students based on the TinyViT 11M architecture, the teacher and the TinyCLIP ViT-8M/16-3M model on MSCOCO text-to-image and image-to-text retrieval tasks. The domain-agnostic students were distilled for one epoch on DataComp medium using either $\mathcal{L}_2$ feature distillation of vision-language distillation based on the CLIP loss. The domain-specific student are subsequently distilled on the synthetic images from the pets domain.

cific domain need to be encoded. We only select query images that contain objects from the specific domains that the students are distilled for and perform retrieval over the entire set of captions. We denote this task as *domain-specific retrieval*. In the following two sections, we discuss the performance of our students in these two tasks and compare them to baselines.

### A.2.1 Domain-Agnostic Retrieval

The results for the domain-agnostic retrieval task can be seen in Figure 5. We report the recall@1,5,10 and 50 for both the text-to-image and text-to-image task. As baselines we include the teacher and the TinyCLIP ViT-8M/16-3M model with the same number of trainable parameters as our TinyViT 11M students. Of our students, we evaluate two models after only domain-agnostic distillation for one epoch on DataComp medium using either $\mathcal{L}_2$ feature distillation or vision-language distillation based on the CLIP loss. In addition, we evaluate two students that were domain- specifically distilled (after initial domain-agnostic distillation) on the synthetic pets or food data. We observe that our domain-agnostic student distilled through feature distillation achieves performances slightly better than the TinyCLIP model. In contrast, the performance of the domain-agnostic student trained through vision-language distillation is substantially worse with only 2% recall@1 in both retrieval tasks. Furthermore, we observe that the retrieval performance of the domain-specific students is worse than for the domain-agnostic students. This can be attributed to the fact that for domain-agnostic retrieval, *all* images have to be encoded and not only images from a specific target domain that the students were distilled for. The evaluation of the domain-specific models on retrieval tasks within their target domains is discussed in the next section.

### A.2.2 Domain-Specific Retrieval

For domain-specific retrieval, we consider image-to-text retrieval and restrict the query images to the subset of MSCOCO that contains objects from specific domains. This task requires only the encoding of images from a restricted number of classes in comparison to domain-agnostic retrieval where all images must be encoded. In Table we report the performance for the pets and food domain. For the pets domain we

| Domain of Query Images | feature distillation $\mathcal{L}_2$ | vision-language distillation CLIP | MP | feature and vision-language distillation combined $\mathcal{L}_2$+CLIP | $\mathcal{L}_2$+MP | Teacher | Domain-Agnostic |
|---|---|---|---|---|---|---|---|
| Pets | **48.7** | 1.2 | 2.0 | 46.1 | 2.0 | 66.2 | 44.9 |
| Food | **47.5** | 2.1 | 1.1 | 40.1 | 7.9 | 62.4 | 41.5 |

Table 6: **Domain-specific retrieval improves image-to-text retrieval in the target domain.** The results state the recall@1 for the domain-specific image-to-text retrieval task on subsets of MSCOCO that correspond to the pets or food domain. Apart from the teacher and domain-agnostic student in the rightmost column, the results are for domain-specific students distilled on synthetic data. **Bold numbers indicate the best performance amongst the students** and red numbers highlight low recall.

select only the images associated to the categories "cat" and "dog" and for the food domain we filter for the categories "banana", "apple", "sandwich", "orange", "broccoli", "carrot", "hot dog", "pizza", "donut" and "cake". We observe that the performance of the domain-specific student trained with feature distillation on synthetic data increases substantially compared the domain-agnostic student. The performance of the students distilled with vision-language distillation drops to a very low level. This demonstrates that our main finding also applies to retrieval: feature distillation on synthetic images improves the performance over pure domain-agnostic distillation while vision-language distillation deteriorates the performance.

### A.3 Details of the Synthetic Data Generation

In this section, we provide further details on the synthetic data generation and the diversification process. As mentioned in Section 3.3, the prompts used to synthesize the images are based on the class names and additional information given by an LLM. For each class, we ask the language model to provide information with respect to four contextual dimensions as well as a superclass. The contextual dimensions are dataset specific and summarized in Table 7. Figure 6 shows a concrete example for a class from the pets dataset. For each of the contextual dimensions we collect 15 or 30 options from Llama 2 7B fine-tuned for chats (Touvron et al., 2023). The larger number of options for the food and ImageNet-100 datasets are used to accommodate its larger test set. In Table 8 we summarize the sizes of the real target datasets. Instead of using all possible combinations of options for the contextual dimensions to generate the prompts, we use combinatorial testing (Ahmed et al., 2017; Nie & Leung, 2011). This approach is inspired by a recent work on systematic error identification (Metzen et al., 2023). It reduces the number of images per class while ensuring that the prompts systematically cover the diversity contained in the answers from the LLM. For example, in case of 15 options per contextual dimension, this results in 265 images per class instead of 50625. The prompts are a comma separated list of the selected options, which are weighted to accommodate for contextual dimensions that are more or less important for certain datasets. These weighted prompts are then used as input for a diffusion model. Specifically, we use Stable Diffusion XL (Podell et al., 2023) LCM LoRA (Luo et al., 2023). To ensure sufficient image quality and diversity we employ a guidance scale of 0.5 and 6 inference steps. Further example images can be found in Figure 7. These also showcase some of the known problems with diffusion models such as parts of the prompts which are missing in the image (Zhang et al., 2023) as in the first example for the food dataset.

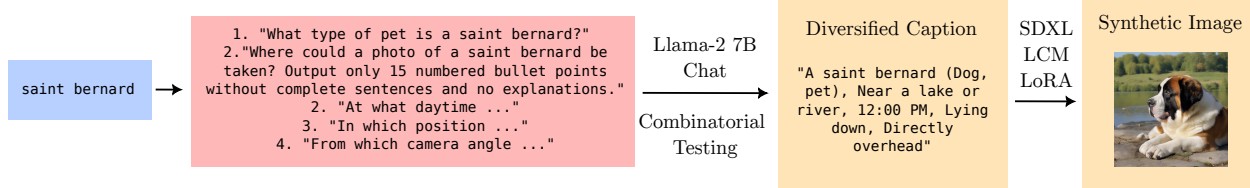

Figure 6: **Process for generating synthetic images.** Illustration for the example class "saint bernard" from the pets dataset.

Oxford Pets

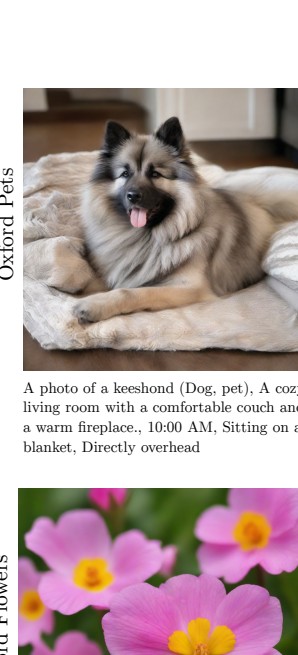 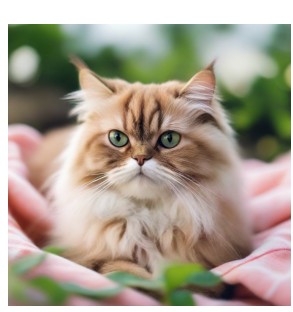 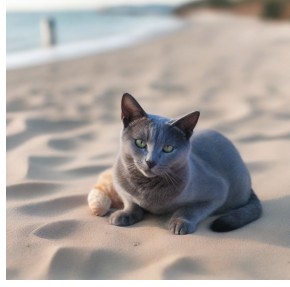

A photo of a keeshond (Dog, pet), A cozy living room with a comfortable couch and a warm fireplace., 10:00 AM, Sitting on a blanket, Directly overhead

A photo of a persian (Cat, pet), A serene garden or park with lush greenery and a comfortable spot for the cat to lounge., Afternoon, around 1-2pm., Lying on a soft blanket, looking adorable, Directly above

A photo of a russian blue (Cat, pet), A picturesque beach with sand and seashells, and a cat-friendly pier., 6:00 PM, Bottom of the frame, Directly above

Oxford Flowers

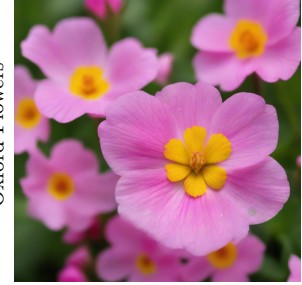 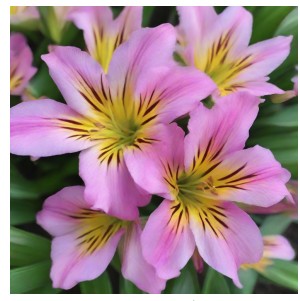 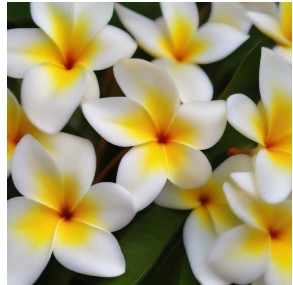

A photo of a pink primrose (Rose, flower), Straight-on, Meadow, Candy Apple, 12pm

A photo of a peruvian lily (Alstroemeria, flower), close-up shot of the blossom, A garden or botanical garden with a variety of flowers and plants., White, Early morning (5:00-6:00 AM)

A photo of a frangipani (Plumeria, flower), Top-down, A jungle or rainforest, , Orange, Early afternoon (1:00-2:00 PM)

Stanford Cars

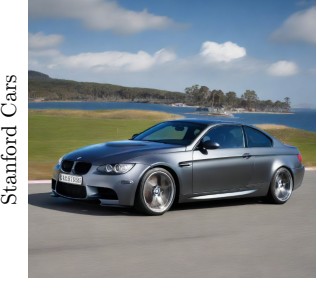 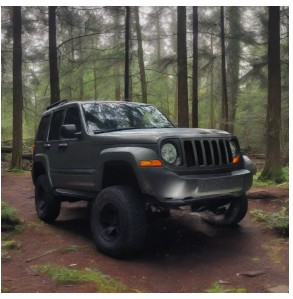 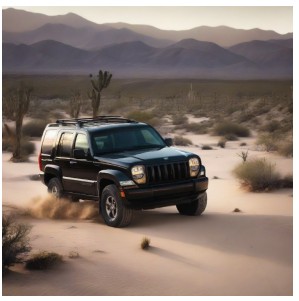

A photo of a Mineral Grey Metallic BMW M3 Coupe 2012 (Sports car, car), a coastal highway with a beautiful ocean view, 2:00 PM, Wide-angle shot from the front

A photo of a Granite Crystal Metallic Jeep Liberty SUV 2012 (Sport, car), A dense forest with tall trees, 2:00 PM, Directly in front of the car

A photo of a Black Jeep Liberty SUV 2012 (Sport, car), A desert landscape with cacti and sand dunes, 6:00 PM, Directly in front of the car

Food 101

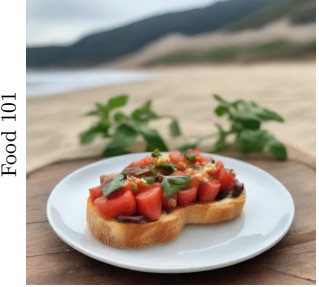 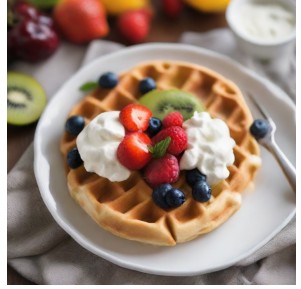 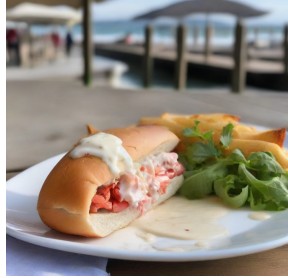

A photo of a bruschetta, as a side dish with grilled meats, Side angle, background is sandy beach with a scenic view, Top-down, Golden hour

A photo of a waffles, with fresh fruit and whipped cream, Side angle, background is a breakfast nook in a cozy kitchen, Top-down angle, 9:00 AM

A photo of a lobster roll sandwich, Served with a side of tangy, creamy sauce., Side angle, background is a picturesque seaside park, Top-down, 2:00 PM

Figure 7: **Examples images.** Taken from the synthetic pets, flowers, cars and food training datasets together with the prompts used to generate them.

| Dataset | Weight of Classname | Contextual Dimensions and Weights in Braket | Options per Dimension | Images per Class |
|---|---|---|---|---|
| Pets | 1.5 | superclass (1.2), locations, position, daytime, camera angle | 15 | 265 |
| Flowers | 1.2 | superclass, color, locations, daytime (0.1), camera angle (0.1) | 15 | 265 |
| Cars | 1.0 | superclass, locations, color, daytime, camera angle | 15 | 265 |
| Food | 1.2 | superclass, locations, way of serving (1.5), daytime(0.1), camera angle(0.1) | 30 | 1011 |
| Textures | 1.5 | superclass (1.2), color, daytime(0.1), camera angle(0.1), location (0.1) | 15 | 265 |
| Aircraft | 1.5 | superclass (0.1), locations, position, daytime, camera angle | 15 | 265 |
| ImageNet-100 | 1.2 | superclass (1.2), locations, position, daytime, camera angle | 30 | 1011 |

Table 7: **Contextual dimensions and prompt weights for the diversified data generation.**

| Dataset | #classes | #training images | #test images |
|---|---|---|---|
| Pets | 37 | 3680 | 3669 |
| Flowers | 102 | 1020 | 6149 |
| Cars | 196 | 8144 | 8041 |
| Food | 101 | 75750 | 25250 |
| Texture | 47 | 1880 | 1880 |
| Aircraft | 100 | 3334 | 3333 |
| ImageNet-100 | 100 | 130000 | 5000 |

Table 8: **Overview over the size of the real target datasets.**

### A.4 Generalization From Real to Synthetic Images.

To consolidate or findings from Section 5, we investigate the domain shift in the opposite direction. That is, we assess how the models trained on real or synthetic images perform on synthetic images. For this purpose, we generate an additional synthetic dataset for pets using the same methodology as for the synthetic training sets. Subsequently, we evaluate the students on this dataset. The results are presented in Table 9. The students distilled through vision-language distillation on real data exhibit lower test accuracies in comparison to feature distillation. For the models trained on synthetic data the reverse is true. Using the CLIP or MP loss results in the highest accuracy on the synthetic test data. As in Section 5 this discrepancy suggests that these models learned features of natural or synthetic images over class-specific features. As a result, their ability to generalize between natural and synthetic images is limited.

### A.5 Top-5 Test Accuracies

To complement the results in Section 5, we provide the Top-5 accuracies of the models. Figure 8 visualizes the results. The trends mirror the observations from the Top-1 accuracy with an even smaller gap to the teacher.

| | | feature distillation | vision-language distillation | | feature and vision-language distillation combined | | | |
|---|---|---|---|---|---|---|---|---|
| Training Data | Test Data | $\mathcal{L}_2$ | CLIP | MP | $\mathcal{L}_2$+CLIP | $\mathcal{L}_2$+MP | Teacher | Domain-Agnostic |
| Real | Synthetic | **91.7** | 85.6 | 82.9 | 89.8 | 86.9 | 93.8 | 79.9 |
| Synthetic | Synthetic | 94.5 | 97.9 | 97.7 | 96.7 | 97.8 | - | - |

Table 9: **Feature distillation generalizes better between real and synthetic images.** The accuracies refer to the domain-specific models for pets that were trained either on real or synthetic images. The evaluation is performed on a synthetic pets dataset which was sampled independently from the train set. Red indicates overfitting to synthetic images.

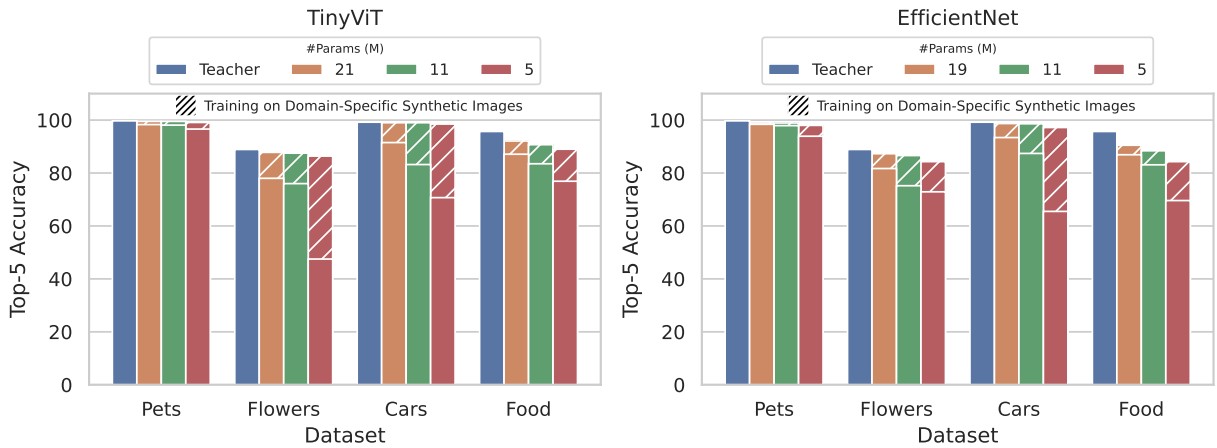

Figure 8: **Top-5 accuracies.** The student were trained on DataComp medium for one epoch (solid) and subsequently on the synthetic datasets for 96 epochs (hatched). The black lines indicate the top-5 accuracy of the teacher.

## A.6 Training Accuracies

In addition to the test accuracies stated in the main paper, we report the training accuracies of the domain-specific models in Table 10. We observe that the models trained with a vision-language loss achieve higher training accuracy than the students obtained from feature distillation. This holds in particular on synthetic data. In combination with the results from Section 5, this underlines our hypothesis that vision-language losses lead to learning datatype specific features over actual class or object specific features.

## A.7 Contrastive Image Loss

As an alternative to the $\mathcal{L}_2$ feature loss, we test a contrastive feature loss that is purely based on the image features of the student and teacher. Using the notation from Section 3.4, it is defined as

$$\mathcal{L}_{\text{contrastive}}^{\text{image}} = \sum_{i=1}^{N} -\log \frac{\exp(\langle \mathbf{I}_i^S, \mathbf{I}_i^T \rangle / \tau)}{\sum_{k=1}^{N} \exp(\langle \mathbf{I}_i^S, \mathbf{I}_k^T \rangle / \tau)} \tag{1}$$

where $\tau$ denotes a learnable temperature parameter. We use this loss both for domain-agnostic and domain-specific distillation of a TinyViT 11M with the same setup as for the $\mathcal{L}_2$ loss in Section 5. The results are reported in Table 11. We observe that for domain-specific distillation, the contrastive image loss results in better performance in comparison to the $\mathcal{L}_2$ loss, while for domain-specific it is the other way around. Yet, the contrastive image loss still clearly outperforms the CLIP loss when training on domain-specific synthetic data. This validates our observation from Section 5 that using a loss purely based on the image features of student and teacher improves the generalization between synthetic and real data.

| Train set | Loss | Pets Top-1 | Pets Top-5 | Flowers Top-1 | Flowers Top-5 | Cars Top-1 | Cars Top-5 | Food Top-1 | Food Top-5 |
|---|---|---|---|---|---|---|---|---|---|
| Real | $\mathcal{L}_2$ | 87.7 | 99.0 | 70.3 | 89.3 | 83.8 | 99.1 | 79.3 | 93.7 |
| | CLIP | 89.7 | 98.5 | 97.3 | 100.0 | 90.6 | 100.0 | 97.6 | 99.2 |
| | MP | 89.1 | 98.5 | 97.5 | 100.0 | 90.7 | 100.0 | 97.5 | 99.2 |
| | $\mathcal{L}_2$+CLIP | 91.2 | 99.9 | 90.6 | 98.0 | 92.2 | 100.0 | 90.8 | 98.3 |
| | $\mathcal{L}_2$+MP | 100.0 | 100.0 | 97.7 | 100.0 | 92.8 | 100.0 | 98.5 | 99.6 |
| Synthetic | $\mathcal{L}_2$ | 95.3 | 100.0 | 68.0 | 90.1 | 75.9 | 95.7 | 84.5 | 96.3 |
| | CLIP | 100.0 | 100.0 | 97.8 | 98.9 | 90.0 | 99.2 | 99.7 | 100.0 |
| | MP | 99.9 | 100.0 | 99.5 | 100.00 | 87.6 | 98.4 | 99.7 | 100.0 |
| | $\mathcal{L}_2$+CLIP | 97.9 | 100.0 | 85.8 | 96.9 | 91.0 | 99.7 | 93.2 | 98.8 |
| | $\mathcal{L}_2$+MP | 100.0 | 100.0 | 99.6 | 100.0 | 94.7 | 100.0 | 99.9 | 100.0 |

Table 10: **Vision-Language distillation yields high training accuracies.** The accuracies refer to a TinyViT 11M student evaluated in the domain-specific datasets it was distilled on. They contrast with the low test accuracies under the domain shift from synthetic to real images in Table 4.

| Train set | Training | Loss | Pets | Flowers | Cars | Food |
|---|---|---|---|---|---|---|
| Real | Domain-Agnostic | Contrastive Image | 72.8 | 39.6 | 46.5 | 54.5 |
| | | Difference to $\mathcal{L}_2$ | +1.4 | +0.5 | +1.5 | +1.4 |
| | | Difference to Contrastive Image-Text (CLIP) | +52.4 | +35.4 | +41.1 | +49.8 |
| | Domain-Specific | Contrastive Image | 83.9 | 64.9 | 81.4 | 80.4 |
| | | Difference to $\mathcal{L}_2$ | -4.8 | -3.5 | -2.4 | -2.6 |
| | | Difference to Contrastive Image-Text (CLIP) | -5.5 | -25.7 | -9.3 | -2.6 |
| Synth. | Domain-Specific | Contrastive Image | 80.5 | 57.7 | 79.3 | 68.8 |
| | | Difference to $\mathcal{L}_2$ | -7.0 | -10.6 | -2.6 | -3.1 |
| | | Difference to Contrastive Image-Text (CLIP) | +13.8 | +18.6 | +15.1 | +40.8 |

Table 11: **Contrastive feature distillation performs comparably to $\mathcal{L}_2$ feature distillation.** Accuracy of the models that were domain-agnostically and domain-specifically trained using distillation with the contrastive image loss. The differences to $\mathcal{L}_2$ feature distillation and training using the CLIP loss are shown in gray.

## A.8 Influence of Image Diversity

To assess the influence of diversified images, we utilize the zero-shot prompts "a photo of a ..." to generate images instead of diversified prompts from a LLM. We sample a synthetic pets dataset with the same number of images per class as in the diversified case. Example images are shown in Figure 9. The diversity of the images decreases, especially with regard to the camera angle, as almost all images show only a frontal shot of the animals with the focus on the face. Furthermore, the variety of backgrounds decreases. We train a TinyViT 11M model on this dataset. The test accuracies on the real pets dataset are stated in Table 12. The accuracy of feature distilled student exhibits only a small decrease in performance in contrast to the diversified images, while the performance of the models which were trained through vision-language distillation decreased significantly. This findings indicates that feature distillation is more robust to a lack of diversity during domain-specific distillation.

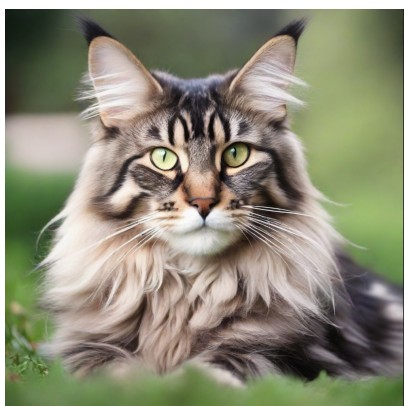 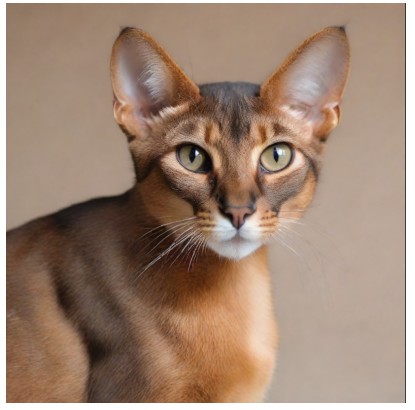 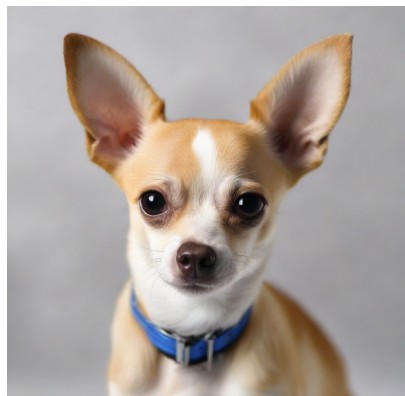

| A photo of a maine coone | A photo of a abyssinian | A photo of a chihuahua |

Figure 9: **Examples for images without diversification.** The shown images were generated using the zero-shot prompts "a photo of ...". The resulting images feature less diversity in comparison to the diversified prompts generated by an LLM depicted in Figure 7.

| Training Data | Test Data | feature distillation $\mathcal{L}_2$ | vision-language distillation CLIP | MP | feature and vision-language distillation combined $\mathcal{L}_2$+CLIP | $\mathcal{L}_2$+MP |
|---|---|---|---|---|---|---|
| Synthetic (simple) | Real | 85.9 | 40.7 | 45.2 | 81.8 | 49.1 |
| Difference to synthetic (diversified) | | -1.6 | -26.0 | -21.3 | -5.4 | -19.0 |
| Synthetic (simple) | Synthetic (diversified) | 93.0 | 86.6 | 85.0 | 93.8 | 87.6 |
| Difference to synthetic (diversified) | | -1.5 | -10.9 | -12.7 | -2.9 | -10.2 |

Table 12: **Feature distillation is more robust to a lack of diversity in the training images.** The reported accuracies refer to TinyViT 11Ms models distilled for one epoch on DataComp medium and subsequently distilled on synthetically generated pets test data. The difference to the results in Section 5, is that the synthetic images were sampled with the zero shot prompts "a photo of ..." instead of diverse prompts from a LLM. The performance of the students distilled through vision-language distillation deteriorates more than that of students distilled through feature distillation.

### A.9 Evaluation Under Common Corruptions

In this section, we provide more details for the performance evaluation under common corruptions. In Table 3, we report the relative performance under corruption (rPC). According to Michaelis et al. (2019), it is defined as

$$rPC = \frac{mPC}{P_{clean}} \quad (2)$$

where $P_{clean}$ is the clean performance and $mPC$ is the mean performance under corruption given by

$$mPC = \frac{1}{N_c N_s} \sum_{c=1}^{N_c} \sum_{s=1}^{N_s} P_{c,s}. \quad (3)$$

$P_{c,s}$ denotes the classification accuracy on test data corrupted with corruption $c$ under severity level $s$. In our case $N_c = 15$ and $N_s = 5$ are the number of corruptions and corruption strengths (Hendrycks & Dietterich, 2019). In addition to Table 3, Table 13 shows the mean performance under corruption and the performance drop compared to the clean performance referred to as the degradation under corruption by Hendrycks & Dietterich (2019).

| Domain-Specific Train Data | Test Data | feature distillation $\mathcal{L}_2$ | vision-language distillation CLIP | MP | feature and vision-language distillation combined $\mathcal{L}_2$+CLIP | $\mathcal{L}_2$+MP | Teacher | Domain-Agnostic |
|---|---|---|---|---|---|---|---|---|
| Real | Real (corrupted) | 72.7 | 68.6 | 70.3 | **78.9** | 71.6 | 76.2 | 43.6 |
|  |  | -16.0 | -19.4 | -18.7 | -12.8 | -19.0 | -13.5 | -27.8 |
| Synthetic | Real (corrupted) | **77.0** | 32.7 | 32.6 | 67.1 | 35.4 | - | - |
|  |  | -10.5 | -34.0 | -34.1 | -20.1 | -22.7 |  |  |
| Real | Synthetic (corrupted) | 86.2 | 76.5 | 72.3 | **87.1** | 79.1 | 89.2 | 77.0 |
|  |  | -5.5 | -9.1 | -10.6 | -2.7 | -7.8 | -4.7 | -16.9 |
| Synthetic | Synthetic (corrupted) | **83.2** | 69.5 | 69.4 | 83.1 | 72.4 | - | - |
|  |  | -10.7 | -28.4 | -28.3 | -16.6 | -25.4 |  |  |

Table 13: **Mean performance under corruption.** The reported accuracies apply to classification on the pets dataset under 15 common corruptions and five severity levels as defined by Michaelis et al. (2019). The numbers in grey state the degradation under corruption.

| Domain-Specific Train Data | Probing Data | feature distillation $\mathcal{L}_2$ | vision-language distillation CLIP | MP | feature and vision-language distillation combined $\mathcal{L}_2$+CLIP | $\mathcal{L}_2$+MP | Teacher | Domain-Agnostic |
|---|---|---|---|---|---|---|---|---|
| Real | Real | 89.8 | 88.3 | 88.9 | 92.1 | 90.2 | 90.4 | 81.6 |
| Real | Synthetic | 82.1 | 87.7 | 88.3 | 87.7 | 88.9 | 84.0 | 71.8 |
| Synthetic | Real | 89.6 | 73.7 | 72.6 | 90.1 | 75.3 | - | - |
| Synthetic | Synthetic | 80.9 | 64.5 | 65.0 | 82.8 | 65.8 | - | - |

Table 14: **Linear probing using real images yields better accuracies than probing with synthetic images.** The reported accuracies are obtained through linear probing of the teacher as well as our domain-specific TinyViT 11Ms models on pets. The linear accuracy of the models that were trained with domain-specific synthetic data is increased substantially by probing with real data compared to probing with synthetic data.

## A.10 Linear Probing

To evaluate the linear probe accuracy of the teacher as well as the TinyViT 11M models on the pets dataset, we fit a linear classifier based on the unnormalized image features after the projection head. The classifier is fitted either using synthetic or real data, where only the case of synthetic data corresponds to the zero-shot setting. The hyperparameter sweeps for the regularization are performed on a validation split as in the original CLIP paper (Radford et al., 2021). The results are shown in Table 14. For the models trained on domain-specific synthetic data, the performance is 8 to 10 % worse when probing with synthetic data in comparison to fitting the linear classification head with real data. This highlights that using linear probing based on real data improves the linear classification accuracy by a substantial margin. In contrast, the true zero-shot linear classifiers, where the classification head is fitted using synthetic data, perform comparable to or worse than pure zero-shot classification using the similarity between the image and prompt embeddings. In contrast to our framework, previous works (Tian et al., 2023b;a; Hammoud et al., 2024) mainly focus on the linear accuracy where the classification head is fitted with real data instead of targeting the true zero-shot setting without any real data which is the more difficult task to accomplish.

## A.11 Linear Classification Head Instead Of CLIP Architecture

Instead of using the CLIP architecture, we train and distill TinyViT 11M model with a linear classification head on the pets dataset for comparison. We either train from scratch or initialized weights from domain-agnostic training on ImageNet-22k (with the exception of the linear classification head which is always randomly initialized). As most of the classes from the pets dataset are contained in ImageNet-22k, the latter does not correspond to a strict zero-shot setting even when subsequently performing domain-specific distillation on synthetic data. To train the models, we optimize the standard cross-entropy loss as well as a sum of cross-entropy loss and the original knowledge distillation loss of Hinton et al. (2015). We use the AdamW optimizer (Loshchilov & Hutter, 2019) with no weight decay and the learning rate is set to $5 \times 10^{-4}$ which is the same as used by Wu et al. (2022) for fine-tuning. First, we observe that the drop in performance when training with synthetic data in comparison to real data is similar to the CLIP models based on vision-

| Domain-Specific Data | Domain-Agnostic Data | CE | CE+KL |
|---|---|---|---|
| Real | - | 47.8 | 47.2 |
| Synthetic | - | 26.9 | 29.9 |
| Real | ImageNet-22k | 91.2 | 91.6 |
| Synthetic | ImageNet-22k | 71.3 | 67.4 |

Table 15: **Students with classification heads exhibit a drop in performance when being trained on synthetic images over natural ones.** Training was performed on the synthetic and real pets datasets for 96 epochs using the cross-entropy loss (CE) with or without knowledge distillation (KL). The architecture is a TinyViT-11M with a linear classification head.

| Domain-Specific Data | Pets | Flowers | Cars | Food |
|---|---|---|---|---|
| Real | 93.4 | 98.8 | 88.9 | 90.6 |
| Synthetic | 86.6 | 67.8 | 53.1 | 58.4 |

Table 16: **Training standard classification models does not yield good zero-shot classifiers.** We fine-tune a ViT-B/32 classification model on either the real or synthetic domain-specific datasets and evaluate the classification accuracies on the real test sets. Training on synthetic images corresponds to the zero-shot setting. We observe a substantial drop in performance between synthetic and real datasets. In particular, the zero-shot performance achieved by training the standard classification model on synthetic images is worse than the ViT-B/32 CLIP teacher and our CLIP students.

language losses. Additionally, the performance of the model with classification head is worse compared to the CLIP models when trained from scratch. In contrast, the classifiers pre-trained on ImageNet-22k and subsequently trained on the domain-specific real training data achieve the best performance overall. This can presumably be attributed to the fact that most of the classes are already included in ImageNet-22k which was used for domain-agnostic distillation. Using knowledge distillation for the models with classification head only has a minor effect.

### A.12 Standard Classification Models as Teachers in the Zero-Sot Setting

An alternative to feature distillation of CLIP models in the zero-shot setting would be the direct distillation of standard classification models with backbone and classification head. Performing feature distillation in this setting requires a teacher with the standard classification architecture. However, since the classification head has to be trained specifically for every set of target classes, there are no general-purpose zero-shot models with this architecture. The only way to use teachers with standard classification architecture in a zero-shot setting is to train them with synthetic images. For this purpose, we consider a ViT-B/32 model that has been pre-trained on ImageNet 21k (Steiner et al., 2022) and train it as a possible zero-shot teacher. In Table 16, we compare the domain-specific training of these models using synthetic and real images. The zero-shot teacher trained with standard classification architecture on synthetic images performs worse than the CLIP teacher and our CLIP students. The gap in accuracy between training on the domain-specific synthetic or real images is up to 35%. This justifies why distilling CLIP models is clearly the better setting for zero-shot classification.

### A.13 Using a Different Teacher

To ablate the role of the teacher model, we distill students from a CLIP ViT B/16 teacher trained on LAION 2B instead of the CLIP ViT B/32 trained on DataComp XL which we used previously. All other factors are left untouched to enable a direct comparison. We perform one epoch of domain-agnostic distillation on DataComp medium followed by domain-specific distillation on four synthetic datasets (pets, flowers, cars, food). The results are shown in Table 17. We find that main finding also holds with a different teacher. Domain-specific distillation through feature distillation results in zero-shot accuracies that closely match the performance of the teacher. In contrast, vision-language distillation using the CLIP loss result in worse accuracies with a difference of up to 79% compared to the teacher.

| | Model | Loss | Training Dataset | Pets | Flowers | Cars | Food |
|---|---|---|---|---|---|---|---|
| CLIP | CLIP ViT-B/16 | CLIP | LAION 2B | 89.3 | 69.7 | 87.0 | 83.2 |
| DA | TinyViT-11M | $\mathcal{L}_2$ | DataComp-medium | 67.0 | 39.2 | 39.0 | 51.7 |
| DA + DS | TinyViT-11M | $\mathcal{L}_2$ CLIP | DataComp-medium + Synthetic | 63.1 | 41.3 | 64.9 | 23.9 |
| | TinyViT-11M | $\mathcal{L}_2$ $\mathcal{L}_2$ | DataComp-medium + Synthetic | 84.1 | 63.2 | 79.5 | 70.4 |

Table 17: **Using a different teacher consolidates our findings.** The setup is the same as for Table 4 but with a CLIP ViT B/16 teacher trained on LAION 2B instead of the CLIP ViT B/32 trained on DataComp XL. "DA" denotes domain-agnostic distillation for one epoch on DataComp medium and "DA+DS" contains the results where synthetic data is used for subsequent domain-specific distillation of the students. The blue box highlights the final models distilled through feature distillation.

### A.14 Using a Different Synthetic Data Generator

To ablate on the role of the synthetic data generator, we perform domain-specific distillation on datasets generated by Stable Diffusion 1.5 LCM LoRA Luo et al. (2023) instead of Stable Diffusion XL LCM LoRA. We keep the setting from Section 5.4 including the diversified prompt generation, the number of inference steps for the data generation as well as the hyperparameters for distilling the students. We select a TinyViT 11M student and perform the domain-specific distillation after domain-agnostic distillation for one epoch on DataComp medium. The results stated in Table 18 confirm our main finding that feature distillation greatly improves the zero-shot classification accuracy over vision-language distillation. In particular, feature distillation is more robust to a change in the synthetic data generator. The drop in performance between Stable Diffusion 1.5 and Stable Diffusion XL LCM lies within 5% for feature distillation. For vision-language distillation the performance deteriorates by up to 23% using the CLIP loss and up to 58% using the MP loss.

| | | feature distillation | vision-language distillation | | feature and vision-language distillation combined | |
|---|---|---|---|---|---|---|
| Synthetic Data Generator | Dataset | $\mathcal{L}_2$ | CLIP | MP | $\mathcal{L}_2$+CLIP | $\mathcal{L}_2$+MP |
| Stable Diffusion 1.5 LCM LoRA | Pets | 86.0 | 50.8 | 48.8 | 83.7 | 51.6 |
| Difference to Stable Diffusion XL LCM LoRA | | -1.5 | -15.9 | -17.7 | -3.4 | -16.5 |
| Stable Diffusion 1.5 LCM LoRA | Flowers | 63.6 | 21.0 | 22.3 | 63.0 | 29.0 |
| Difference to Stable Diffusion XL LCM LoRA | | -4.6 | -18.1 | -20.3 | -5.0 | -20.4 |
| Stable Diffusion 1.5 LCM LoRA | Cars | 78.9 | 33.2 | 11.9 | 76.7 | 63.1 |
| Difference to Stable Diffusion XL LCM LoRA | | -3.0 | -23.8 | -58.1 | -6.3 | -19.3 |
| Stable Diffusion 1.5 LCM LoRA | Food | 70.0 | 16.0 | 18.0 | 67.3 | 29.6 |
| Difference to Stable Diffusion XL LCM LoRA | | -2.0 | -12.0 | -5.2 | -3.2 | -12.4 |

Table 18: **Feature distillation is more robust to a change in synthetic image generator.** The reported accuracies are for TinyViT 11Ms student distilled for one epoch on DataComp medium and subsequently distilled on synthetically generated pets test data. The difference to Section 5 is that the synthetic images were generated by Stable Diffusion 1.5 LCM LoRA instead of Stable Diffusion XL LCM LoRA. We find that feature distillation is more robust to the change in diffusion model and the accuracies decrease substantially less in comparison to vision-language distillation.

### A.15 Multi-Positive Contrastive Loss

To adapt the multi-positive (MP) loss to our setting, we replace the anchor sample by the embedding of a class-specific zero-shot prompt. By $\mathbf{Z}_k$ denote the normalized embedding of the zero-shot prompt for class $k$ and by $\mathbf{I}_i$ the normalized embedding of image $i$. The image label is given by $l(\mathbf{I}_i)$. Given class $k$ from a set of $K$ classes and batchsize $N$, the contrastive distribution is given by

$$q_i(k) = \frac{\exp(\langle \mathbf{I}_i, \mathbf{Z}_k \rangle / \tau)}{\sum_{j=1}^{K} \exp(\langle \mathbf{I}_i, \mathbf{Z}_j \rangle / \tau)} \tag{4}$$

and the ground-truth categorical distribution is

$$p_i(k) = \frac{\mathbb{1}_{l(\mathbf{I}_i)=k}}{\sum_{j=1}^{N} \mathbb{1}_{l(\mathbf{I}_j)=k}}. \tag{5}$$

The overall MP loss is then computed as

$$\mathcal{L}_{\mathrm{MP}} = -\frac{1}{K} \sum_{k=1}^{K} \sum_{i=1}^{N} p_i(k) \log q_i(k). \tag{6}$$

### A.16 Theoretical Bound on Teacher-Student Agreement

Conclusively, we present a theoretical motivation of the $\mathcal{L}_2$ feature loss for the distillation of a CLIP image encoder. Therefore, we consider the following notions. Let be $\mathcal{D}_{train} = \{(i_i, t_i) | i \in [N_{train}]$ be a train set consisting of image-caption pairs and $\mathcal{D}_{test} = \{(i_i, l_i) | i \in [N_{test}]\}$ a test set with image-classlabel pairs. By $\mathbf{I}^T(i)$ denote the normalized embedding of image $i$ from the teacher image encoder and by $\mathbf{I}^S(i)$ the normalized embedding of image $i$ from the student image encoder. Similarly, let $\mathbf{Z}_k$ be normalized embedding of the zero-shot prompt "a photo of a {classname}" of class $k \in [N_{\mathrm{classes}}]$. In this setting, the following statement holds.

**Lemma 1.** *Assume that for a given $\epsilon > 0$ it holds that for every image $i \in \mathcal{D}_{test}$ there exists an image $\tilde{i} \in \mathcal{D}_{train}$ such that*

$$\|\boldsymbol{I}^T(i) - \boldsymbol{I}^T(\tilde{i})\|_2 < \epsilon \text{ and } \|\boldsymbol{I}^S(i) - \boldsymbol{I}^S(\tilde{i})\|_2 < \epsilon \tag{7}$$

*If it holds that $\|\boldsymbol{I}^T(\tilde{i}) - \boldsymbol{I}^S(\tilde{i})\|_2 < \epsilon$ for every image $\tilde{i} \in \mathcal{D}_{train}$, then*

$$\langle \boldsymbol{I}^S(i), \boldsymbol{Z}_{p_S(i)} \rangle - \langle \boldsymbol{I}^S(i), \boldsymbol{Z}_{p_T(i)} \rangle < 6\epsilon \tag{8}$$

*for every image $i \in \mathcal{D}_{test}$ where $p_S(i)$ and $p_T(i)$ are the teacher and student class predictions.*

$$p_T(i) = argmax_{k \in [N_{classes}]} \langle \boldsymbol{I}^T(i), \boldsymbol{Z}_k \rangle \text{ and } p_S(i) = argmax_{k \in [N_{classes}]} \langle \boldsymbol{I}^S(i), \boldsymbol{Z}_k \rangle \tag{9}$$

*Proof.* Without loss of generality, we assume there exists an image $i \in \mathcal{D}_{test}$ and $j, k \in [N_{\mathrm{classes}}], j \neq k$ such that for the student

$$\langle \mathbf{I}^S(i), \mathbf{Z}_j \rangle > \langle \mathbf{I}^S(i), \mathbf{Z}_k \rangle \tag{10}$$

but for the teacher

$$\langle \mathbf{I}^T(i), \mathbf{Z}_j \rangle < \langle \mathbf{I}^T(i), \mathbf{Z}_k \rangle. \tag{11}$$

Then, it holds that

$$\langle \mathbf{I}^S(i), \mathbf{Z}_j \rangle = \langle \mathbf{I}^S(i) - \mathbf{I}^T(i), \mathbf{Z}_j \rangle + \langle \mathbf{I}^T(i), \mathbf{Z}_j \rangle \tag{12}$$

$$< \langle \mathbf{I}^S(i) - \mathbf{I}^T(i), \mathbf{Z}_j \rangle + \langle \mathbf{I}^T(i), \mathbf{Z}_k \rangle \tag{13}$$

$$\leq \|\mathbf{I}^S(i) - \mathbf{I}^T(i)\|_2 \|\mathbf{Z}_j\|_2 + \langle \mathbf{I}^T(i), \mathbf{Z}_k \rangle \tag{14}$$

$$= \|\mathbf{I}^S(i) - \mathbf{I}^T(i)\|_2 + \langle \mathbf{I}^T(i), \mathbf{Z}_k \rangle \tag{15}$$

$$= \|\mathbf{I}^S(i) - \mathbf{I}^T(i)\|_2 + \langle \mathbf{I}^T(i) - \mathbf{I}^S(i), \mathbf{Z}_k \rangle + \langle \mathbf{I}^S(i), \mathbf{Z}_k \rangle \tag{16}$$

$$\leq 2\|\mathbf{I}^S(i) - \mathbf{I}^T(i)\|_2 + \langle \mathbf{I}^S(i), \mathbf{Z}_k \rangle. \tag{17}$$

By using assumption 7, there exists an image $\tilde{i} \in \mathcal{D}_{train}$ such that we get

$$\|\mathbf{I}^S(i) - \mathbf{I}^T(i)\|_2 \leq \|\mathbf{I}^S(i) - \mathbf{I}^T(\tilde{i})\|_2 + \|\mathbf{I}^T(\tilde{i}) - \mathbf{I}^T(i)\|_2 \tag{18}$$

$$\leq \|\mathbf{I}^S(i) - \mathbf{I}^S(\tilde{i})\|_2 + \|\mathbf{I}^S(\tilde{i}) - \mathbf{I}^T(\tilde{i})\|_2 + \|\mathbf{I}^T(\tilde{i}) - \mathbf{I}^T(i)\|_2 \tag{19}$$

$$\leq 3\epsilon \tag{20}$$

Overall, we get

$$\langle \mathbf{I}^S(i), \mathbf{Z}_j \rangle - \langle \mathbf{I}^S(i), \mathbf{Z}_k \rangle < 2\|\mathbf{I}^S(i) - \mathbf{I}^T(i)\|_2 \leq 6\epsilon \tag{21}$$

which proves the statement.

$\square$

Lemma 1 highlights that minimizing the $\mathcal{L}_2$ loss on a training set where the image embeddings are close to the embeddings of the test images yields a sufficient criterion for agreement of the teacher and student on the test set. This motivates directly optimizing the $\mathcal{L}_2$ loss through feature distillation over vision-language distillation.

