# OpenReview forum: "Feature Distillation Improves Zero-Shot Transfer from Synthetic Images"
_TMLR — Accepted by TMLR_

### Review · Reviewer_tYTw · 2024-08-23

**Summary Of Contributions:**

This paper presents a framework for fine-tuning vision language models for zero-shot domain-specific tasks through feature distillation. Specifically this paper focuses on image classification tasks with known classes (domain) but no access to downstream data. The fine-tuning data is generated through prompting image generation models. This paper presents results showing that distilling features, instead of end-to-end fine-tuning of vision-language models, is more robust to spurious features in the dataset. At the same time, initial domain-agnostic feature distillation is also important for achieving best downstream performance. The resulted models present competitive performances when compared with models trained on domain-specific data.

**Audience:**

Yes

**Broader Impact Concerns:**

Should add a discussion about if feature distillation would retain biases in the teacher model or image generation model.

**Claims And Evidence:**

Yes

**Requested Changes:**

check for typos, two things I noticed:

abstract missing a word:  "Instead, we investigate the use **of** synthetic images for this purpose."

finding 5 (bottom of page 8): " ... Therefore, we compare domain-agnostic distillation on DataComp medium and ImageNet as well as no domain-specific distillation..." -> I think the "domain-specific" should also be "domain-agnostic"

**Strengths And Weaknesses:**

Strengths:
- This paper investigates an important problem of how to efficiently distill large models into more efficient smaller models for domain specific tasks.
- The proposed method approaches feature distillation in a low-cost manner through leveraging synthetic data.
- This paper examines different design components/factors that influences overall performances, including pre-training and model sizes.
- This paper is well organized and clearly presented. The experiments cleanly support the claimed findings.

Weaknesses:
- The scope of this work is a little limited: solely focuses on image classification as downstream task and only experimented with CLIP
- It is interesting to see that domain-agnostic pretraining boosts finetuning, however, it would be useful to know if the benefit come from the pretraining data being real images or simply more diverse; an ablation could be adding domain-agnostic pretraining with synthetic data that is generated to represent a diverse range of classes

Overall the claims and experimental results in this paper is convincing

---

> ### Author Response · Authors · 2024-09-05
> **Author feedback**
>
> We thank the reviewer for the constructive feedback and recognizing that our paper investigates an important problem through convincing findings that are cleanly supported by experiments. In the following we discuss the suggested improvements.
>
> **Comment 1: The scope of this work is a little limited: solely focuses on image classification as downstream task and only experimented with CLIP.**
>
> We appreciate the reviewer's feedback that our paper's evaluation of the models is focused solely on classification. To address this, we verify our claims on an additional task: retrieval on the MSCOCO dataset. For the corresponding results, please see our global comment and the detailed version in Section A.2 of our revised submission. By evaluating our distilled CLIP image encoders on both classification and retrieval tasks, we assessed the same downstream tasks as existing works on distilling CLIP models (MobileCLIP, TinyCLIP, and CLIP-KD). We would like to emphasize that it is a common paradigm in the literature to solely evaluate CLIP models on classification and retrieval and not a limitation specific to our work. This can be attributed to the fact that additional downstream tasks such as captioning require additional architectural components and go beyond evaluation only the CLIP model.
>
> For experiments with other architectures than CLIP we refer to our answer for comment 3 mentioned by reviewer 9mpk. There, we explain why we use CLIP models as teachers and distilling standard classification models with backbone and classification head is inferior to feature distillation of CLIP image encoders.
>
> **Comment 2: It would be useful to know if the benefit comes from the pretraining data being real images or simply more diverse.**
>
> We thank the reviewer for bringing up this very interesting point! To investigate the role of natural vs synthetic images during domain-agnostic distillation, we perform domain agnostic distillation on SynthCI dataset from SynthCLIP by Hammoud et al. which consists of 30 million images from Stable Diffusion. Using the same distillation setup as for DataComp and ImageNet in finding 3, the results are as follows. All models are trained using feature distillation.
>
> | Training Data | Pets | Flowers | Cars | Food |
> | ------------- | ---- | ------- | ---- | ---- |
> | Domain-agnostic distillation on **SnythCI 30M** | 80.9 | 46.2 | 8.4  | 48.2 |
> | Domain-agnostic distillation on **DataComp medium** | 71.4 | 39.9 | 45.0 | 52.9 |
> | Domain-agnostic distillation on **SnythCI 30M** + Domain-specific distillation on synthetic datasets | 87.8 | 69.5 | 81.3 | 73.4 |
> | Domain-agnostic distillation on **DataComp medium** + Domain-specific distillation on synthetic datasets | 87.5 | 68.3 | 81.9 | 71.9 |
>
> We observe that the difference of the final performance (after domain-specific distillation) using domain agnostic-distillation on real (DataComp) vs synthetic images (SynthCI) is smaller than 2% on all four datasets. After pure domain-agnostic distillation, the performance of the models trained on the SynthCI datasets are comparable to the model trained on DataComp medium, apart from the cars dataset, where the SynthCI model only achieves 8.4% accuracy. These observations indicate that the crucial part for domain-agnostic distillation is the data being diverse and not being real. We added this statement to finding 3.
>
> **Comment 3: Typos**
>
> We thank the reviewer for pointing out the typos and have corrected them for the revised version.

---

### Review · Reviewer_9mpk · 2024-08-24

**Summary Of Contributions:**

This paper introduces a zero-shot domain-specific distillation scheme for classification tasks, where student models are first distilled on domain-agnostic datasets. The synthetic data used for distillation is generated by models, enabling the zero-shot approach. The authors present findings that offer insights into improving performance.

**Audience:**

Yes

**Claims And Evidence:**

Yes

**Requested Changes:**

I suggest the authors to add explainations and related experiments to address the issues and concerns mentioned in the 'Strengths And Weaknesses' part.

**Strengths And Weaknesses:**

The pipeline is sound, and the proposed method achieves good performance in the experimental results. However, I have concerns about the findings and the setup of this paper. In the experiment related to Finding 1, the authors add artificial spurious features to the images. While this can demonstrate that such spurious features can lead to worse performance in vision-language distillation, it does not prove that the actual decline in performance in real-world data is due to these spurious features. This is a sufficient but not necessary condition.

Additionally, this paper only performs distillation for classification tasks, which is just a small part of the CLIP model's capabilities. This raises the question of whether the method proposed in the paper is truly a general-purpose domain-specific distillation method, and whether it is really necessary to distill a model from CLIP instead of directly training a smaller model for classification.

---

> ### Author Response · Authors · 2024-09-05
> **Author feedback part 1**
>
> We thank the reviewer for the constructive feedback and recognizing that we introduce a sound framework, and our students achieve good performances. In the following, we discuss the suggested improvements.
>
> **Comment 1: Observing a performance drop under artificial spurious features "does not prove that the actual decline in performance in real-world data is due to these spurious features. This is a sufficient but not necessary condition.**
>
> We fully agree with the reviewer that robustness to spurious features is a sufficient but not necessary condition. However, the goal of finding 1 is not to claim that the robustness to spurious features is a necessary condition to bridge the decline in performance in real-world data. Being a necessary condition would indicate that only models that are robust to spurious features can bridge the performance gap. This is a very strong claim and potentially not true. In general, we do not claim anywhere that the list of our findings is complete or that any of the individual findings are themselves necessary. Instead, our paper should be read such that feature distillation has a set of favorable properties – finding 1 focuses on spurious features but findings 2 and 3 are equally important for our framework - that jointly improve the transfer between synthetic and real images. We made this clearer by weakening our title and replaced “enables” with “improves”. Furthermore, we have revised the limitations section and explicitly point out that there might be other approaches besides feature distillation to improve the transfer performance between synthetic and real images. We point out that both reviewers tYTw  and PziH agree with our claims and explicitly state that the findings are “clearly stated, precisely and without hyperbole” as well as “convincing”.
>
> **Comment 2: This paper only performs distillation for classification tasks, which is just a small part of the CLIP model's capabilities. This raises the question of whether the method proposed in the paper is truly a general-purpose domain-specific distillation method.**
>
> We acknowledge the reviewer's comment that the evaluation of our models in the paper is limited to classification. As an additional task we consider retrieval on MSCOCO. For the results we refer to our global comment and the supplementary material of the revised submission. With the evaluation on classification and retrieval we now evaluated our distilled CLIP image encoders on the same tasks as all previous works for distilling CLIP models (MobileCLIP, TinyCLIP and CLIP-KD).

---

> > ### Author Response · Authors · 2024-09-05
> > **Author feedback part 2**
> >
> > **Comment 3: Is it really necessary to distill a model from CLIP instead of directly training a smaller model for classification?**
> >
> > We thank the review for pointing out this interesting question. In the following, we demonstrate that "directly training a smaller model for classification" does not yield competitive results in a zero-shot setting. Directly training/distilling smaller models in a zero-shot setting can be performed in two ways. In the following, when we speak of “standard classification models” we refer to the architecture with backbone and classification head.
> >
> > *Option 1: Distilling a standard classifier student from a standard classifier teacher*
> >
> > Performing feature distillation of standard classification models with classification heads requires standard teachers with classification heads. However, there exist no general-purpose teachers with this standard classification architecture that can be used for zero-shot classification like CLIP, as the classification head must be trained to match the domain-specific target classes. In a zero-shot setting, the only direct option is to use synthetic data to train the model. For this purpose, we fine-tune a ViT-B classification model initialized with ImageNet-21k pre-trained weights on the domain-specific target datasets. This results in the following accuracies:
> >
> > | Model | Domain-Specific Training Data | Pets Accuracy | Flowers Accuracy | Cars Accuracy | Food Accuracy |
> > | ----------------------------- | ----------------------------- | ------------- | ---------------- | ------------- | ------------- |
> > | ViT-B/32 + classification head | Real                          | 93.4          | 98.8             | 88.9          | 90.6          |
> > | ViT-B/32 + classification head | Synthetic                     | 86.6          | 67.8             | 53.1          | 58.4          |
> > | CLIP ViT-B/32                   | -                             | 89.7          | 72.9             | 85.4          | 82.9          |
> > | Domain-specific students distilled from CLIP ViT-B/32 using feature distillation on synthetic data | Synthetic | 87.5 | 68.3 | 81.9 | 71.9 |
> >
> > We observe that the zero-shot accuracies of these models are worse than both the CLIP teacher and our models trained through feature distillation using synthetic data. Thus, we conclude that (feature) distillation with standard classification models as teachers is inferior in a zero-shot setting.
> >
> > *Option 2: Distilling a standard classifier student from a CLIP teacher*
> >
> > The second option is distilling standard classification models (students) from CLIP teachers through logit-based distillation. To test this setup, we use the domain-specific synthetic datasets and distill models that are initialized with pre-trained weights from ImageNet-21k. As students we use TinyViT 11M models together with a linear classification head. This results in the following accuracies.
> >
> > | Domain-Specific Training Data | Pets Accuracy | Flowers Accuracy | Cars Accuracy | Food Accuracy |
> > | ----------------------------- | ------------- | ---------------- | ------------- | ------------- |
> > | Real                           | 91.6          | 96.3             | 91.7          | 90.2          |
> > | Synthetic                      | 67.4          | 41.5             | 65.4          | 24.7          |
> >
> > We observe that this setup suffers from a similarly large gap in performance between training on synthetic images and evaluating on natural images as vision-language distillation. From this observation, we conclude that distilling standard classification models from CLIP teachers through logit-based distillation is inferior to feature distillation in a zero-shot setting.
> >
> > *Related Option: Fitting classification heads through linear probing.*
> > A related approach for training a "direct" classification model is to fit a linear classification head through linear probing on top of the distilled CLIP image encoders. For the details on linear probing, we refer to Section A.11 of our submission. The crucial point is that when training the linear classification head through linear probing with synthetic images instead of real images, the performance deteriorates as well.
> >
> > Overall, none of the three options for "directly" distilling classification models reach the performance of the image encoder distilled feature distillation and zero-shot evaluated like CLIP models. This justifies that directly training a smaller model for classification is inferior to the setup in our paper.

---

> ### Author Response · Authors · 2024-09-12
> **Clarification of Comment 1**
>
> After reading your review again we are not sure whether we interpreted your comment "while this can demonstrate that such spurious features can lead to worse performance in vision-language distillation, it does not prove that the actual decline in performance in real-world data is due to these spurious features" the right way. We understood it such that you wanted to highlight finding 1 individually might not be necessary to improve the zero-shot transfer from synthetic data. We updated our submission to clarify that feature distillation has several favorable properties investigated in our findings that jointly improve the zero-shot transfer (see our initial response) and there could be alternative approaches with different properties that achieve similar results. In case you have any more specific doubts, we would be happy about further clarification.

---

### Review · Reviewer_PziH · 2024-08-29

**Summary Of Contributions:**

This paper studies zero-shot distillation, i.e. transferring domain knowledge from a large teacher foundation model to a smaller, domain-specific student model, yet without having access to images from the target domain (only relevant text prompts). Specifically, it considers distilling from a large CLIP model to smaller, domain-specific version, using synthetic data, and targeting the task of image classification. Overall the paper is analytic rather than proposing any innovative model components; instead it defines a pipeline and measures the benefit of certain design choices within this. The experimental results show that using a feature-space loss for distillation works better than the vision-language losses (including standard CLIP loss) that are more common in this setting, and also that the resulting model exhibits better robustness to input perturbations.

**Audience:**

Yes

**Broader Impact Concerns:**

None -- this work does not raise any significant concerns (nor is there any broader impact statement present)

**Claims And Evidence:**

Yes

**Requested Changes:**

- revise the title to be weaker, or somehow justify saying "enables"
- consider swapping the results sections so 5.4 comes first – the discussion of feature robustness (particularly with rather arbitrary ellipses overlaid) etc. is less interesting (in my view) than the 'headline' results of 5.4
- some details from A.2 maybe should be in the main text (at least that it's SDXL with a very small step-count) – there is currently an almost complete lack of detail about synthetic data generation in the main paper
- p8, penultimate line, "as well as no domain-specific distillation" – should this be no domain-*agnostic* distillation? it seems to contradict the following parenthesised text

**Strengths And Weaknesses:**

Strengths
- The task of zero-shot distillation (which is rather unusual) is sufficiently well motivated
- The proposed framework is novel; although it does not contain any specifically new technical components, it allows controlled study of how different factors (explored less systematically in earlier works, or applied to different task settings) affect student model performance in the zero-shot distillation setting
- Detailed experiments are conducted to measure the impacts of various different factors, using two different student models (a CNN and a ViT), and several image classification datasets.
- Quantitative results show that feature distillation significantly improves on using just the vision-language loss, and leads to state-of-the-art performance for small CLIP-style models.
- Moreover, the feature-distilled models are found to be more robust to various image perturbations.
- The findings and contributions are clearly stated, precisely and without hyperbole; they include specific actionable insights that will be useful to other practitioners interested in this task
- Overall the text is clear and readable; the paper is very well structured; the figures are appropriate

Weaknesses
- The only task considered is image classification; this is a relatively uncommon use-case for CLIP-style models, and it would have been nice to see some results on other tasks like captioning or segmentation.
- The core finding that feature distillation is useful, is not particularly surprising given its success in other distillation tasks / settings
- The title is a bit strong – while the text is generally moderate and precise in its claims, to say that feature distillation **enables** 0-shot transfer is an over-statement; more accurate to say it **greatly improves** it, given the results in Table 4
- All experiments use a single teacher model – there is no analysis of how well the findings hold up with a different teacher architecture, or a different pretraining dataset
- All experiments use a single synthetic data generator (a variant of SDXL) – there is no analysis of what impact this choice has on the findings. The argument that synthesis is preferred over retrieval is also somewhat weak – the retrieval source dataset does not need to be exactly the same dataset as used for training, just similarly distributed

---

> ### Author Response · Authors · 2024-09-05
> **Author feedback part 1**
>
> We thank the reviewer for the constructive feedback and recognizing that we investigate a well-motivated problem with precisely stated findings and present them in a well-structured manner. In the following, we discuss the comments for improving our submission.
>
> **Comment 1: The only task considered is image classification.**
>
> We appreciate the reviewer's feedback that the evaluations in our paper were limited to classification. In response, we have included an additional task of retrieval on the MSCOCO dataset to verify our claims on an additional downstream task. For more details, please refer to our global comment and the details in Section A.2 of our revised submission. By evaluating our distilled CLIP image encoders on both classification and retrieval tasks, we consider the same downstream tasks as existing works on distilling CLIP models (MobileCLIP, TinyCLIP, and CLIP-KD). It is important to note that evaluating CLIP models solely on classification and retrieval is a common practice in the literature and not a limitation specific to our work. This can be attributed to the fact that further downstream tasks such as captioning require additional architectural components and go beyond evaluating only the CLIP model.
>
> **Comment 2: The core finding that feature distillation is useful, is not particularly surprising given its success in other distillation tasks / settings.**
>
> Given the findings from our submission and previous papers, it does indeed seem like feature distillation is a widely applicable technique that is beneficial in many settings. However, the combination of distillation and synthetic data is a novel setting that has not been explored in any previous works. Therefore, the influence of feature distillation on this task has not been analyzed before. Given the substantial effort that has been taken by the community to train models on synthetic data without fully bridging to the performance on real data, we think that our insights into why feature distillation is effective in this setting are highly useful (and given the simplicity of the approach also somewhat surprising).
>
> **Comment 3:  The title is a bit strong.**
>
> We acknowledge the reviewer’s assessment and have changed the title to “Feature Distillation Improves Zero-Shot Transfer from Synthetic Images”. Relevant sections in the main text have been adjusted as well.
>
> **Comment 4:  All experiments use a single teacher model.**
>
> To ablate the role of the teacher model, we distill TinyViT 11M students from a CLIP ViT B/16 teacher trained on LAION 2B instead of the CLIP ViT B/32 trained on DataComp XL which we used throughout the paper. All other factors are left untouched to enable a direct comparison. We perform one epoch of domain-agnostic distillation on DataComp medium followed by domain-specific distillation on four synthetic datasets (pets, flowers, cars, food). This yields the following results:
>
> | Model | Accuracy on Pets | Accuracy on Flowers | Accuracy on Cars | Accuracy on Food |
> | -------- | ---------------- | ----------------- | ---------------- | --------------- |
> | CLIP ViT B/16 LAION 2B (teacher) | 89.3 | 69.7 | 87.0 | 83.2 |
> | TinyViT 11M student after domain-agnostic distillation on DataComp | 67.0 | 39.2 | 39.0 | 51.7 |
> | TinyViT 11M student after domain-agnostic distillation on DataComp and subsequent domain-specific distillation on synthetic data using **vision-language distillation** with the CLIP loss | 63.1 | 41.3 | 64.9 | 23.9 |
> | TinyViT 11M student after domain-agnostic distillation on DataComp and subsequent domain-specific distillation on synthetic data using L2 **feature distillation** | 84.1 | 63.2 | 79.5 | 70.4 |
>
> We find that our observations from the paper also hold in this setting. Domain-specific distillation through feature distillation results in zero-shot accuracies that closely match the performance of the teacher. In contrast, vision-language distillation using the CLIP loss result in worse accuracies with a difference of up to 79% compared to the teacher.

---

> > ### Author Response · Authors · 2024-09-05
> > **Author feedback part 2**
> >
> > **Comment 5: All experiments use a single synthetic data generator.**
> >
> > As an alternative to StableDiffusion XL LCM LoRA, which we use as the synthetic data generator for the results in the submission, we perform experiments using StableDiffusion 1.5 LCM LoRA for the pets, flowers, cars and food domains. In the following table, we compare these two settings for feature distillation using the L2 loss and vision-language distillation using the CLIP loss. The results state the zero-shot classification accuracies.
> >
> > | Synthetic Data Generator | Distillation loss | Pets | Flowers | Cars | Food |
> > | ------------------------ | ----------------- | ---- | ------- | ---- | ---- |
> > | TinyViT 11M after domain-agnostic distillation on DataComp and subsequent domain-specific distillation on synthetic data from **StableDiffusion 1.5 LCM LoRA** | L2 (**feature distillation**) | 86.0 | 63.6 | 78.9 | 70.0 |
> > | Performance drop compared to **StableDiffusion XL LCM LoRA** (as used in the paper) |  | -1.5 | -4.6 | -3.0 | -2.0 |
> > | TinyViT 11M after domain-agnostic distillation on DataComp and subsequent domain-specific distillation on synthetic data from **StableDiffusion 1.5 LCM LoRA** | CLIP loss (**vision-language distillation**) | 50.8 | 21.1 | 33.2 | 16.0 |
> > | Performance drop compared to **StableDiffusion XL LCM LoRA** (as used in the paper) |  | -15.9 | -18.1 | -28.3 | -12.0 |
> >
> > We emphasize that our main observation from the paper also holds with a different synthetic data generator. Distilling the students through feature distillation on the synthetic images again yields strong zero-shot performance close to the teacher, while vision-language distillation on synthetic data deteriorates the performance. For feature distillation, the performance drop between StableDiffusion XL and StableDiffusion 1.5 lies within a margin of 5% while with vision-language distillation the performance is up to 24% worse with StableDiffusion 1.5. For the detailed results, we refer to Section A14 of the revised submission.
> >
> > **Comment 6: The argument that synthesis is preferred over retrieval is also somewhat weak.**
> >
> > While the diffusion models that we use were trained on publicly available datasets, several state-of-the-art image generators like DALL-E or Flux were trained on closed-source datasets. In these cases, retrieval from the underlying datasets is not possible. In practice, an additional motivation for using synthetic images over retrieval might be restrictive licenses of large-scale datasets for retrieval or generally the availability of suitable datasets for certain specific domains.
> >
> > **Comment 7: Consider swapping the results sections so 5.4 comes first.**
> >
> > We agree with the reviewer that the headline results in Section 5.4 are presented rather late in the paper.  However, we think that first discussing the properties that influence the main results makes Section 5 much easier to follow and understand. To include the ‘headline’ results also earlier in the paper, we have highlighted the zero-shot accuracies on the 6 target datasets of the students trained through feature distillation or vision-language distillation in figure 1. Like this, the main results are already apparent in the introduction.
> >
> > **Comment 8: Some details from A.2 maybe should be in the main text.**
> >
> > We added more details to the main text including the guiding scale and number of inference steps for the diffusion models. The fact that the step count for the stable diffusion models is so small is due to the fact that we use LCM LoRA models which are specifically trained to generate high-quality images with 4-6 inference steps. In our setup, the generation process for one image is almost 4 times faster compared to the latent diffusion models used in Stabel Diffusion, which is why we chose these models.
> >
> > **Comment 9: Typos.**
> >
> > We thank the reviewer for pointing out the typo and have corrected it for the revised submission.

---

### Author Response · Authors · 2024-09-05
**Additional Downstream Task: Evaluation on Retrieval**

We acknowledge the fact that the evaluation in our paper was focused only on classification. Thus, we evaluate the student on the retrieval task on MSCOCO. We consider two settings. First, we evaluate the text-to-image and image-to-text Recall@1. The aim of text-to-image retrieval is to select images from a large pool that best match a given caption (the so-called query). For image-to-text retrieval, the setup is the other way around, i.e. querying with an image and selecting relevant captions. With CLIP models, text-to-image retrieval is performed by selecting the images or captions from the pool whose embeddings have the highest cosine similarity to the query. This task requires strong domain-agnostic performance as all images from the validation set of MSCOCO, which range across various domains, must be encoded in a meaningful manner. Thus, we compare two of our domain-agnostic students (TinyViT 11M) distilled for one epoch DataComp medium using either L2 feature distillation or vision-language distillation based on the CLIP loss. As baselines we include the CLIP ViT/B teacher and a TinyCLIP ViT-8M/16-3M model with the same number of trainable parameters as our students. The following table states the results.


| Recall@1 | Teacher (baseline) | TinyCLIP (baseline) | Domain-agnostic student trained with feature distillation (L2 loss) | Domain-agnostic student trained with vision-language distillation (CLIP loss) |
| -------- | ------------------ | ------------------- | ------------------ | ------------------ |
| Text-to-Image | 36.2% | 22.4% | 26.3% | 2.2% |
| Image-to-Text | 54.5% | 36.8% | 39.6% | 2.1% |


We observe that our domain-agnostic student distilled through feature distillation achieves performances slightly better than the TinyCLIP model. In contrast, the performance of the domain-agnostic student trained through vision-language distillation is substantially worse with only 2% Recall@1 in both retrieval tasks.

Next, we evaluate domain-specific students on the retrieval task. These students are distilled to match the teacher on specific domains. Therefore, we consider only the image-to-text retrieval task and restrict the images to the subset of MSCOCO that shows objects from a specific domain. This task only requires the image encodings from the specific domains that the students were distilled for. For example, for the domain-specific students for Oxford Pets we select only the images associated to the categories "cat" and "dog". This results in the following image-to-text retrieval performance. The domain-specific students were distilled on synthetic images.


| Recall@1 | Domain-specific student trained with the CLIP loss (vision-language distillation) | Domain-specific student trained with the L2 loss (feature distillation) | Domain-agnostic student trained for one epoch on DataComp medium with feature distillation (L2 loss) |
| -------- | -------------------------------------------------------------------------------- | ---------------------------------------------------------------------- | --------------------------------------------------------------------------------------------------- |
| Image-to-text retrieval with only “cat” and “dog” images | 1.7% | **48.71%** | 44.9% |


We observe that the performance of the domain-specific student trained with feature distillation (on synthetic data) increases over the domain-agnostic student. In contrast, the performance of the students distilled with vision-language distillation drops to a very low level. This demonstrates that **our main finding also applies to retrieval**: feature distillation on synthetic images improves the performance over pure domain-agnostic distillation while vision-language distillation deteriorates the performance. A more in-depth evaluation of the retrieval task can be found in Section A.2.

The evaluation on tasks such as image captioning or segmentation models goes beyond the scope of purely distilling CLIP models as additional architectural components are involved. An example for such architectures are LLava models which can be used for image captioning and are built on CLIP image encoders. However, only one to four percent of the number of parameters of the entire model belong to the image encoder. Thus, the remaining architectural components must also be considered for meaningful model compression. This exceeds the scope of our work.

Lastly, we would like to emphasize that it is a common paradigm in the literature to solely evaluate classification and retrieval. Existing works on distilling CLIP models (MobileCLIP, TinyCLIP, CLIP-KD) only evaluate on classification and retrieval too. This can again be attributed to the fact that additional downstream tasks go beyond evaluation of the CLIP model itself.

---

### Author Response · Authors · 2024-09-12
**Summary of the Discussion**

We thank the reviewers for their constructive feedback and suggestions to improve the submission. At the end of the official discussion period, we provide an overview of the main comments from the reviewers together with the improvements.



**Strengths**
- **Well-motivated and relevant research question.** In our submission we investigate the distillation of image encoders in a zero-shot setting using synthetic data. This task is "well motivated" (Reviewer PziH) and "important" (Reviewer tYTw ).

- **Sound experimental setup.** For our experiments we distill CLIP image encoders into task-specific students. In this setup, we investigate the influence of the loss (using feature distillation over vision-language distillation) on the performance of the students. The pipeline is "sound" (Reviewer 9mpk) and the "experiments cleanly support the claimed findings" (Reviewer tYTw).

- **Strong performance.** The students distilled with feature distillation achieve state-of-the-art performance (Reviewer PziH, Reviewer 9mpk).

- **Convincing claims.** Our findings regarding the influence of spurious features, common corruptions and domain-agnostic distillation are "clearly stated, precisely and without hyperbole" (Reviewer PziH) and "convincing" (Reviewer tYTw).

**Improvements**

- **Additional Downstream Task.** Our initial submission evaluated the performance of the student image encoders solely on image classification. To complement this, we considered retrieval as an additional downstream task. We emphasize that our main result also holds for this task: feature distillation improves the performance over vision-language distillation in the zero-shot setting. On domain-agnostic retrieval on MSCOCO, training a student with L2 feature distillation yields 39.6%(image-to-text)/26.3%(text-to-image) recall@1 while vision-language distillation using the CLIP loss results in only 2.2%(image-to-text)/2.1%(text-to-image). For domain-specific image-to-text retrieval, the difference is even greater. The student distilled with feature distillation on the pets domain achieves 48.7% recall@1 in contrast to only 1.7% for vision-language distillation.

    The evaluation of the performance of distilled CLIP models on classification and retrieval is the common paradigm in literature. Existing         works on the distillation of CLIP models (MobileCLIP, TinyCLIP, CLIP-KD) also only evaluate classification and retrieval. Tasks such as image labeling or segmentation require additional architectural components apart from CLIP image encoders. Distilling such models goes beyond the scope of our submission as we focus on CLIP image encoders.

- **Ablations.** We added several ablations regarding the dataset for domain-agnostic distillation (Reviewer tYTw), the architecture of the student (Reviewer 9mpk), the synthetic data generator and the teacher (Reviewer PziH). All additional experiments confirm our findings.

-  **Clarification of the Scope.** Reviewer PziH pointed out that the title is a bit too strong and Reviewer 9mpk asked for clarification regarding the necessity of the individual findings. In response, we adjusted our title and relevant sections of the manuscript, changing the phrase "feature distillation enables the zero-shot transfer" to "feature distillation improves the zero-shot transfer." Additionally, we expanded the limitations section to explicitly state that feature distillation is sufficient to enhance the performance of students trained on synthetic data, but it is not necessarily the only approach to yield such improvements.

---

### Decision · Action_Editor_DAKK · 2024-10-18

**Recommendation:** Accept as is

**Comment:**

Besides the above strengths, the authors have provided comprehensive responses to address the concerns raised in the first version,  including additional downstream tasks such as retrieval (beyond classification), and the results further bolster the paper’s claims. This demonstrated the transferability of the method to tasks beyond classification, making it more versatile. While some reviewers noted that the paper focuses only on classification, the additional experiments on retrieval and detailed ablation studies strengthen the findings. The authors also clarify the limitations, acknowledging that feature distillation is not the only method for improving zero-shot transfer but is highly effective in this context. The reviewers appreciated the precise and non-hyperbolic presentation of the claims. The revised title and expanded sections addressing the concerns make the contributions clearer and more appropriately scoped.

**Audience:**

Broad interests.

**Claims And Evidence:**

The paper has generally positive reviews, with all the reviewers leaning toward accepting the work. Reviewers are satisfied with the following evident strengths:
1. Robustness and Transferrable Performance: It demonstrates that using feature distillation (minimizing the distance between student and teacher image features) without image captioning enhances robustness against spurious features and data corruptions, significantly improving the transfer performance from synthetic to real images.

2. Strong Empirical Results: The distilled student models achieve zero-shot performance comparable to the much larger teacher models (ViT-B/32) on fine-grained classification datasets, despite using 92% fewer parameters. This is validated across several tasks, including image-to-text retrieval.

3. Domain-Specific and Domain-Agnostic Distillation: The paper showcases that the feature-distilled models perform better than those distilled using vision-language approaches in domain-specific and domain-agnostic settings.